# Statistical context dictates the relationship between feedback-related EEG signals and learning

Matthew R Nassar[1,2]*, Rasmus Bruckner[3,4,5], Michael J Frank[1,6]

[1]Robert J. & Nancy D. Carney Institute for Brain Science, Brown University, Providence, United States; [2]Department of Neuroscience, Brown University, Providence, United States; [3]Department of Education and Psychology, Freie Universität Berlin, Berlin, Germany; [4]Center for Lifespan Psychology, Max Planck Institute for Human Development, Berlin, Germany; [5]International Max Planck Research School on the Life Course (LIFE), Berlin, Germany; [6]Department of Cognitive, Linguistic, and Psychological Sciences, Brown University, Providence, United States

**Abstract** Learning should be adjusted according to the surprise associated with observed outcomes but calibrated according to statistical context. For example, when occasional changepoints are expected, surprising outcomes should be weighted heavily to speed learning. In contrast, when uninformative outliers are expected to occur occasionally, surprising outcomes should be less influential. Here we dissociate surprising outcomes from the degree to which they demand learning using a predictive inference task and computational modeling. We show that the P300, a stimulus-locked electrophysiological response previously associated with adjustments in learning behavior, does so conditionally on the source of surprise. Larger P300 signals predicted greater learning in a changing context, but less learning in a context where surprise was indicative of a one-off outlier (oddball). Our results suggest that the P300 provides a surprise signal that is interpreted by downstream learning processes differentially according to statistical context in order to appropriately calibrate learning across complex environments.
DOI: https://doi.org/10.7554/eLife.46975.001

*For correspondence:
mattnassar@gmail.com

## Introduction

People are capable of rationally adjusting the degree to which they incorporate new information into their beliefs about the world (*Behrens et al., 2007*; *Nassar et al., 2010*; *Cheadle et al., 2014*; *d'Acremont and Bossaerts, 2016*; *Diederen et al., 2016*). In environments that include discontinuous changes (changepoints) normative learning requires increasing learning when beliefs are uncertain or when observations are most surprising (*Nassar et al., 2010*; *Nassar et al., 2012*). Human participants display both of these tendencies, albeit to varying degrees (*Nassar et al., 2010*; *Nassar et al., 2012*; *Nassar et al., 2016*).

A major open question in the learning domain is how the brain achieves such apparent adjustments in learning rate. This question has fueled a number of recent studies that have identified neural correlates of surprise in functional magnetic resonance imaging (fMRI) (*McGuire et al., 2014*), electroencephalography (EEG) (*Jepma et al., 2016*; *Jepma et al., 2018*), and pupil signals (*Nassar et al., 2012*) that predict subsequent learning behavior. These signals might reflect candidate mechanisms for a general system to adjust learning rate (*Behrens et al., 2007*; *O'Reilly et al., 2013*; *Iglesias et al., 2013*), yet the generality has yet to be established outside of discontinuously changing environments, where surprise and learning are tightly coupled.

The relationship between surprise and learning is complex and depends critically on the overarching statistical context. We refer to learning as the degree to which an observed prediction error promotes measurable behavioral updating. While changing environments require increased learning in the face of surprising information, stable environments with outliers ('oddballs'), dictate less learning from surprising information (*d'Acremont and Bossaerts, 2016*). People are capable of this type of robust learning rate adjustment that deemphasizes surprising information (*Cheadle et al., 2014*; *d'Acremont and Bossaerts, 2016*; *Summerfield and Tsetsos, 2015*), yet the learning signals measured under such conditions do not correspond directly to those observed in changing environments. Most notably, a number of candidate learning signals measured through fMRI do not reflect learning rate when considering a broader set of statistical contexts (*d'Acremont and Bossaerts, 2016*).

However, prior studies on EEG correlates of learning seem to favor the idea that a late, stimulus-locked positivity referred to as the P300, tracks learning in a broader range of statistical contexts. While the central parietal component of the P300 (P3b) has been long known to reflect surprise (*Mars et al., 2008*; *Kolossa et al., 2015*; *Kopp et al., 2016*; *Seer et al., 2016*; *Kolossa et al., 2012*), recent work suggests it relates to learning (*Fischer and Ullsperger, 2013*) even after controlling for the degree of surprise in changing environments (*Jepma et al., 2016*; *Jepma et al., 2018*). In a stationary environment where integration of sequential samples is required to make a subsequent decision, a late posterior positivity, reminiscent of the P300, predicts the degree to which a particular sample influences the subsequent decision (*Wyart et al., 2012*). Interestingly, within this particular task, more surprising outcomes tended to exert less influence on decisions (*Cheadle et al., 2014*; *Summerfield and Tsetsos, 2015*), suggesting that this late positivity might provide a general learning or updating signal, irrespective of statistical context. This idea would be in line with a prominent theory of P3b function, which emphasizes its role in updating context representations – sometimes defined in terms of items stored in working memory (*Donchin, 1981*; *Donchin and Coles, 1988*; *Polich, 2003*; *Polich, 2007*).

Here we tested the idea that the P3b provides a general learning signal that is independent of the statistical context. In particular, we measured learning behavior using a modified predictive inference task and a normative learning model and examined how learning behavior and surprise related to evoked potentials measured through EEG. We found that people are capable of contextually adjusting learning in response to surprise: they tended to learn more from surprising outcomes when those outcomes were indicative of changepoints, but learned less from surprising outcomes when those outcomes were indicative of an oddball. Outcome evoked potentials reminiscent of a parietal P300 were related to surprising events irrespective of context. The magnitude of this P300 response on a given trial positively predicted learning in the presence of changepoints, but negatively predicted learning in the presence of oddballs. This conditional relationship between the P300 signal and learning was most pronounced in individuals who showed the largest behavioral adjustments in the two conditions. Furthermore, early P300 signaling predicted subsequent learning even when controlling for variability in learning behavior that could be explained by the best behavioral model.

Taken together these findings suggest that the P300 does not naively reflect increased behavioral updating, but may play a role in adaptively increasing or decreasing learning in response to surprising information, depending on the statistical context.

## Results

We used EEG to measure electrophysiological signatures of feedback processing while participants performed a modified predictive inference task (*Nassar et al., 2010*) designed to dissociate surprise from learning. Predictions were made in the context of a video game that required participants to place a shield at a location on a circle in order to block cannonballs that would be fired from a cannon located at the center of the circle (*Figure 1A*). Surprise and learning were manipulated independently using two different task conditions. In the *oddball* condition, the aim of the cannon drifted slowly from one trial to the next (*Figure 1B*, dotted line) and cannonball locations were distributed around the point of cannon aim (*Figure 1B*, green points nearby dotted line) or, occasionally and unpredictably, uniformly distributed around the circle (*oddballs*; see green point on trial 11 of *Figure 1B* for example). In the *changepoint* condition, the cannon aim remained constant for an

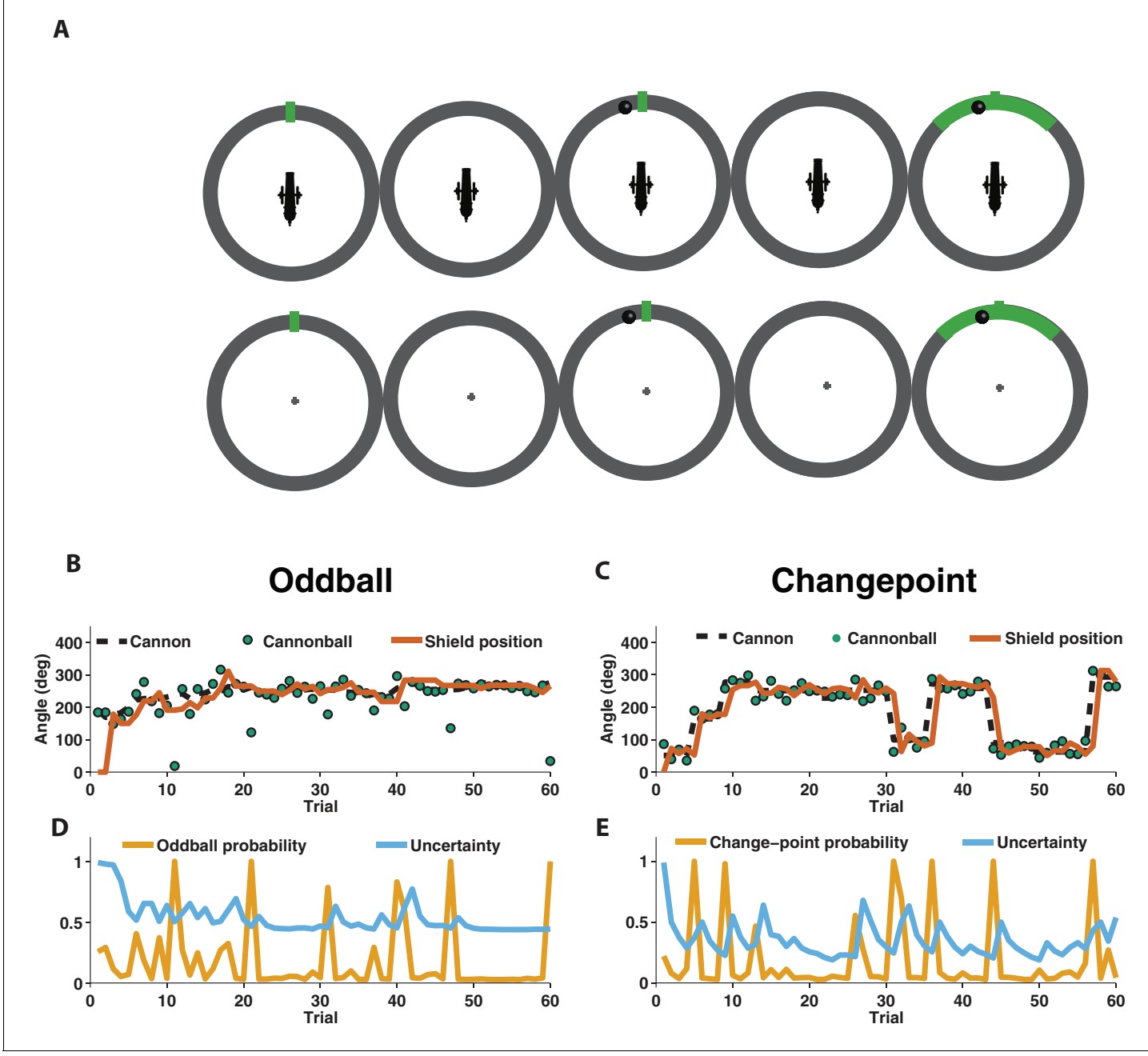

**Figure 1.** Measuring learning in different statistical contexts with a predictive inference task. (A) Participants were trained to place the center of a shield (green tick; prediction phase [left]) at the aim location of a cannon (training task [top]) in order to block a cannonball shot from it (outcome phase [top middle]) with a shield that varied in size from trial to trial and was revealed at the end of the trial (shield phase [top right]). After training, participants were asked to complete the same task, but without a visual depiction of the cannon, which required them to infer the aim of the cannon based on the sequence of previously observed cannonballs (test task [bottom]). (B) In oddball blocks, cannon aim (dotted black line) followed a random walk and cannonball locations were typically drawn from a Von Mises distribution centered on the true cannon aim (green points), but were occasionally drawn from a uniform distribution across the entire circle (oddball trials). Participants placed their shield on each trial (brown line) providing information about their inference about the cannon aim. (C) In changepoint blocks, cannon aim was stationary for most trials but was occasionally resampled uniformly from possible angles (changepoint) and cannonball locations were always drawn from a Von Mises distribution centered on the true cannon aim (green points). (D and E) Optimal inference could be approximated in both generative environments by tracking and adjusting learning according to relative uncertainty and the probability of an unlikely event (oddball or changepoint).

DOI: https://doi.org/10.7554/eLife.46975.002

unpredictable duration, and was then re-aimed at a new location on the circle at random (*change-points*; *Figure 1C*, dotted line). Cannonball locations were always distributed around the point of cannon aim in this condition (*Figure 1C*, green points).

## Behavior of human participants and normative model

In both conditions, participants were instructed to place a shield on each trial in order to maximize the chances of blocking the upcoming cannonball (*Figure 1B and C*, orange line). However, behavior differed qualitatively in these two conditions, which can be observed clearly in the example participant data in *Figure 1*. In particular, shield placements were not updated in response to extreme outcomes in the oddball condition (oddballs; *Figure 1B*) but were updated dramatically in response to extreme outcomes in the changepoint condition (changepoints; *Figure 1C*).

To quantitatively analyze the differences between the two task conditions, we extended a previously developed normative learning model (*Nassar et al., 2010*; *Nassar et al., 2016*). The model approximates optimal inference using an error-driven learning rule by adjusting learning from trial to trial according to two latent variables. The first latent variable tracks the probability with which the most recent outcome was generated from an unexpected generative process (oddball probability in *Figure 1D*; changepoint probability in *Figure 1E*), whereas the second latent variable tracks the model's uncertainty about the true cannon aim (*Figure 1D and E*; uncertainty). Critically, the model stipulates that surprising events in the oddball condition, which are tracked through the model's estimate of oddball probability, should reduce learning, as oddballs are unrelated to future cannonball locations (*d'Acremont and Bossaerts, 2016*). In contrast, the model stipulates that surprising events in the changepoint condition, which are tracked through the model's estimate of changepoint probability, should amplify learning, as changepoints render prior cannonballs (and thus prior beliefs) irrelevant to the problem of predicting future ones (*Adams and MacKay, 2007*; *Wilson et al., 2010*). Qualitatively, behavior from the example participant seems to follow these prescriptions, with adjustments in shield position fairly minimal on trials that include a spike in oddball probability (*Figure 1B,D*), but fairly large on trials that include a spike in changepoint probability (*Figure 1C,E*).

The normative model also makes quantitative prescriptions for how learning should be adjusted according to surprise differentially in the changepoint and oddball conditions. The surprise of a given outcome can be measured crudely through the degree to which a cannonball location differed from that which was predicted (e.g., the shield position). Larger absolute prediction errors indicate a higher degree of surprise, and higher oddball or changepoint probabilities depending on the task condition. Learning in this task can be measured through the degree to which a participant adjusts the shield position in response to a given prediction error (*Nassar et al., 2010*), and a fixed rate of learning would correspond to a straight line mapping each prediction error onto a corresponding shield update, where the slope of the line can be thought of as the *learning rate* (*Figure 2C*, gray lines). The normative learning model does not prescribe a fixed learning rate across all levels of surprise; instead it prescribes higher learning rates for more surprising outcomes in the changepoint condition (*Figure 2C*, orange) and lower learning rates for more surprising outcomes in the oddball condition (*Figure 2C*, blue).

Participants adjusted learning behavior in accordance with normative predictions, albeit with considerable heterogeneity across trials and participants. Shield updating behavior and corresponding prediction errors for an example participant reveal the basic trend predicted by the normative model, although exact updates were variable from one trial to the next (*Figure 2D*). To summarize the degree to which updating behavior of individual subjects was contingent on key task variables, we constructed a linear regression model that described trial-by-trial updates in terms of prediction errors as well as key task variables thought to modulate the degree to which prediction errors are translated into updates (*Figure 2E*) including condition (changepoint versus oddball block), surprise (as measured by changepoint or oddball probability estimates from normative model), and their multiplicative interaction (capturing the degree to which learning is increased for surprising outcomes in the changepoint context, but decreased for surprising outcomes in the oddball context). As expected, prediction error coefficients were positive, capturing a tendency for participants to update shield position toward the most recent cannonball position (*Figure 2F*, red; mean/SEM beta = 0.58/0.04, t = 14.4, dof = 38, p=$6\times10^{-17}$). Furthermore, participants systematically adjusted the degree to which they did so according to condition (*Figure 2F*, green; mean/SEM beta = 0.08/ 0.02, t = 3.1, dof = 38, p=0.003), but not significantly according to surprise (*Figure 1F*, blue; mean/

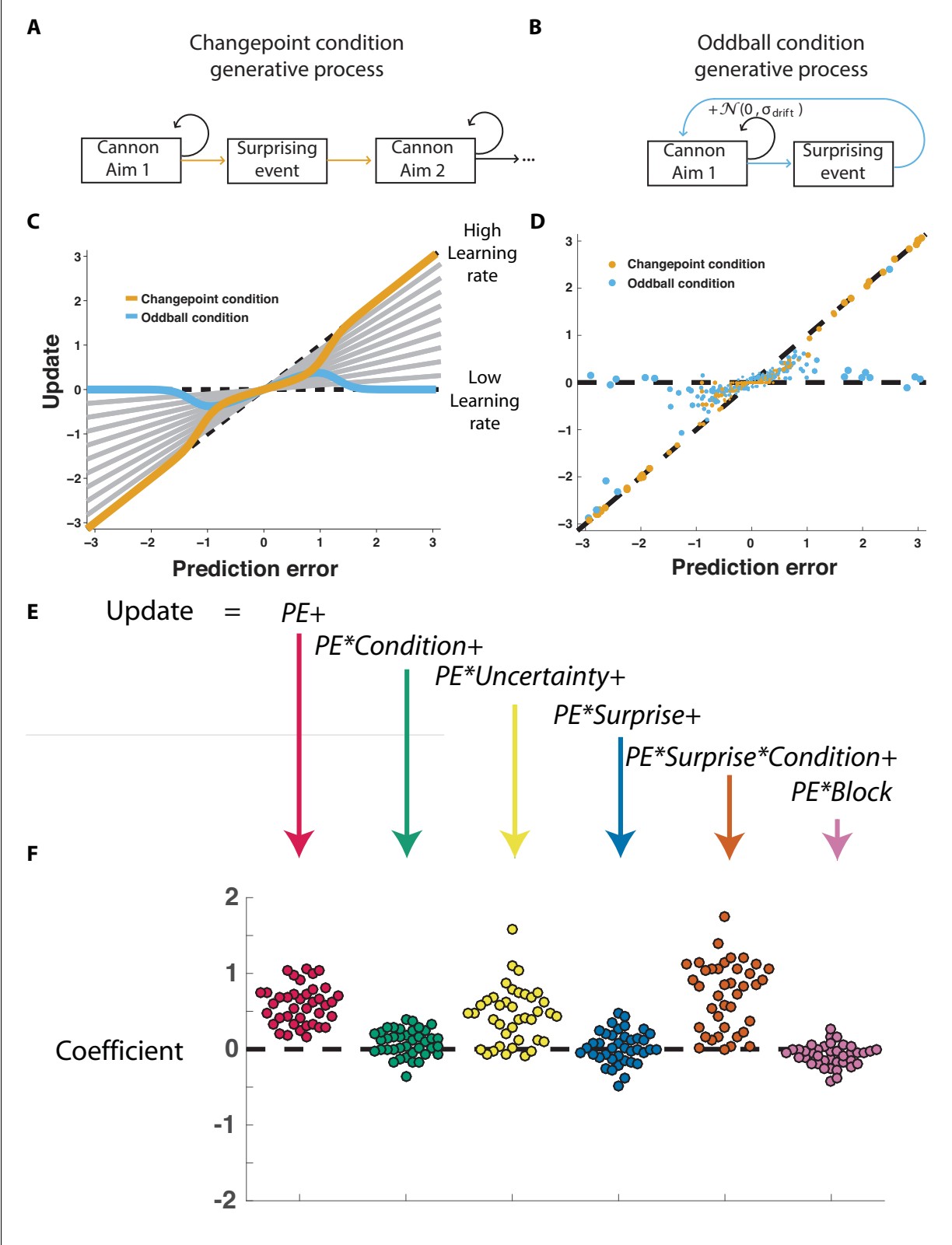

**Figure 2.** Participants scale learning according to surprise differently in changepoint and oddball contexts as would be expected for normative learning rate adjustment. (A) In the changepoint condition, surprising events (changepoints) signaled a transition in the aim of the cannon whereas (B) in the oddball condition, surprising events (oddballs) were unrelated to the process through which the aim of the cannon transitioned. (C) Learning rate in the cannon task can be described by the slope of the relationship between prediction error (signed distance between cannonball and shield; abscissa) and

*Figure 2 continued on next page*

*Figure 2 continued*

update (signed change in shield position after observing new cannonball location; ordinate). Fixed learning rate updating corresponds to a line in this space whose slope is uniform across prediction errors and reflects the learning rate (gray lines). In contrast, normative learning dictates that the slope should decrease for extreme prediction errors in the oddball condition (blue) but increase for extreme prediction errors in the changepoint condition (orange). (D) Prediction error (abscissa) and update (ordinate) for each trial (points) in each condition (designated by color) completed by a single example participant. Size of points is inversely related to density of data for improved visualization. (E) Trial updates for each subject were fit with a regression model that included prediction errors (to measure fixed learning rate) as well as several interaction terms to assess how learning depended on various factors. (F) Coefficients from regression model fit to individual subjects (points) revealed an overall tendency to update toward recent cannonball locations (red, t = 14.4, dof = 38, p=$10^{-17}$), and a tendency to do so more in the changepoint condition (green, t = 3.1, dof = 38, p=0.003), when uncertain (yellow, t = 7.5, dof = 38, p=$4\times10^{-9}$), and on trials where the cannonball was not blocked by the shield (pink, t = −3.4, dof = 38, p=0.001). The model revealed that there was no consistent effect of surprise on learning across both conditions (blue, t = 0.8, dof = 38, p=0.43), but that there was a strong interaction between surprise and condition (orange, t = 9.9, dof = 38, p=$4\times10^{-12}$) whereby surprise tended to increase learning in the changepoint condition but decrease learning in the oddball condition.

DOI: https://doi.org/10.7554/eLife.46975.003

The following figure supplement is available for figure 2:

**Figure supplement 1.** Behavioral interaction was driven by positive effects of surprise on learning in the changepoint condition and negative effects of surprise on learning in the oddball condition.
DOI: https://doi.org/10.7554/eLife.46975.004

SEM beta = 0.03/0.03, t = 0.8, dof = 38, p=0.43). Critically, surprise robustly impacted learning in opposite directions for the two conditions, as indicated by the interaction between surprise and condition (*Figure 2F*, orange; mean/SEM beta = 0.71/0.07, t = 9.9, dof = 38, p=$4\times10^{-12}$). Specifically, positive coefficients indicate that sensitivity to prediction errors was increased for surprising outcomes in the changepoint condition and decreased for surprising outcomes in the oddball condition (*Figure 2—figure supplement 1*), as predicted by the normative model.

## Electrophysiological signatures of feedback processing

We took a data driven approach to identify electrophysiological signatures of feedback processing. First we regressed feedback-locked EEG data collected simultaneously with task performance onto an explanatory matrix that included separate binary variables reflecting changepoint and oddball trials (as opposed to neutral trials that did not involve a rare event), amongst other terms (*Figure 3A*, left). Spatiotemporal maps for changepoint and oddball coefficients were combined to create a *surprise* contrast (changepoint +oddball) and a *learning* contrast (changepoint − oddball) for each subject. Contrasts were aggregated across subjects to create a map of t-statistics (*Figure 3A*, right), and spatiotemporal clusters of electrode/timepoints exceeding a cluster-forming threshold were tested against a permutation distribution of cluster mass to spatially and temporally organized fluctuations in voltage that related to task variables.

When applied to the *surprise* contrast, this procedure yielded a number of significant clusters distributed across electrodes and timepoints (*Figure 3C*). One cluster of positive coefficients spanning 300–700 ms after onset of the cannonball location was of particular interest, given its consistency with the timing and direction of the canonical P300 response. Examining the spatial distribution of coefficients during this period revealed an early frontocentral locus of positive coefficients (350 ms; *Figure 3B*, left) that moves posterior and eventually dissipates over the subsequent 350 ms (*Figure 3B*, middle and right). The positive *surprise* contrast within the cluster included positive contributions of both changepoint and oddball trials (*Figure 3—figure supplement 1*).

The time course of positive *surprise* coefficients (peak t-statistic = 390 ms) is consistent with a P300 response locked to the outcome (cannonball location). Furthermore, the dynamics with which the positivity moves from anterior to posterior central electrodes is reminiscent of a transition from P3a to P3b signaling, with the spatial profile of early time points (e.g., 350 ms) consistent with the frontal P3a and the spatial profile of later time points more consistent with the parietal P3b (e.g., 500 ms). Average outcome-locked event related potentials in a frontocentral electrode (FCz) reveal a positive deflection from 300 to 500 ms (*Figure 3D*, black). This deflection is enhanced on both changepoint and oddball trials (*Figure 3D,E*, orange and blue), reminiscent of the P3a component, also referred to as the novelty P300. Posterior electrode (Pz) event-related potentials (ERPs) reveal a later and longer lasting positive deflection in response to a new outcome (*Figure 3F*, black). This

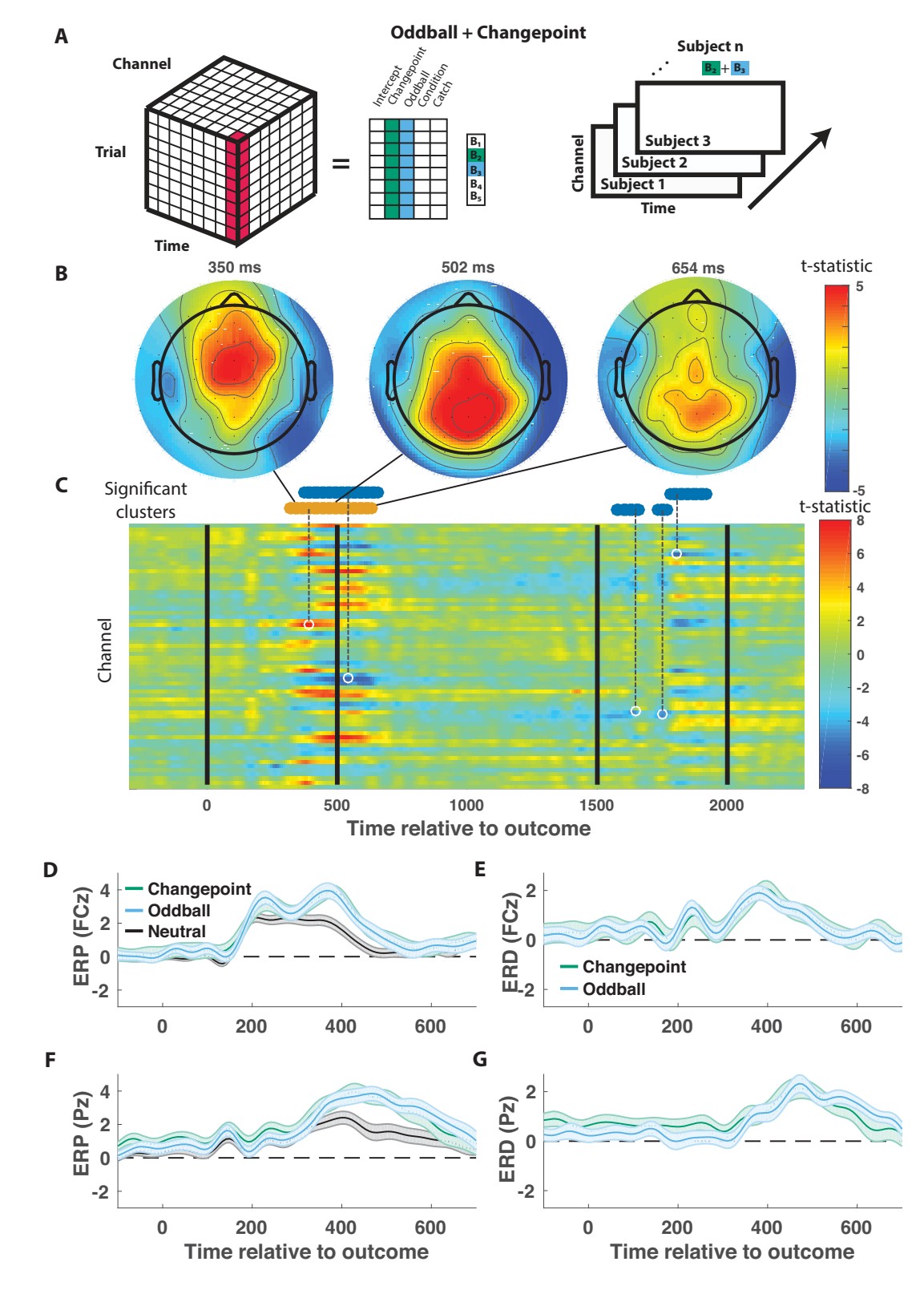

**Figure 3.** Outcome-locked central positivity reflects surprise irrespective of context. (A) Trial-series of EEG data for a given electrode and timepoint was regressed onto an explanatory matrix that contained separate binary regressors for changepoint and oddball trials (left). A t-statistic map was created for each electrode and time point on the surprise coefficient contrast (right). (B and C) T-statistic map for surprise contrast across time (abscissa; C) and channel (ordinate; C) along with corresponding topoplots (B). The time course of spatiotemporal clusters that survived multiple

*Figure 3 continued on next page*

*Figure 3 continued*

comparisons correction via permutation testing is depicted above heat plot (orange indicates positive cluster, blue indicates negative). The time and channel corresponding to the maximum absolute t-statistic for each such cluster are depicted with a white circle and connected to their respective cluster with a dashed line (**C**). (**D and F**) Mean/SEM (line/shading) event related potentials (ERP, microvolts) sorted by trial type (orange = changepoint, blue = oddball, black = other trials) for frontocentral (**D**; FZc) and central posterior (**F**; Pz) electrodes. (**E and G**) Mean/SEM (line/shading) event related difference (ERD) waveforms computed by subtracting the ERP for 'neutral' trials (eg. trials where outcome emerged from the expected generative process) from the average ERP for changepoint and oddball trials at frontocentral (**E**; FZc) and central posterior (**G**; Pz) electrodes.

DOI: https://doi.org/10.7554/eLife.46975.005

The following figure supplements are available for figure 3:

**Figure supplement 1.** P300 spatiotemporal cluster reflected surprise in both changepoint and oddball conditions.

DOI: https://doi.org/10.7554/eLife.46975.006

**Figure supplement 2.** The difference between changepoint and oddball events was not reflected by any event related EEG signal.

DOI: https://doi.org/10.7554/eLife.46975.007

positive deflection is enhanced on both changepoint and oddball trials (*Figure 3F,G*, orange and blue), reminiscent of the P3b, or updating component of the P300. Since the spatial and temporal profiles of this cluster were consistent with what has been referred to in previous literature as the P300, we will refer to it as a P300 signal.

In contrast to the EEG signature of *surprise*, which included a robust and extended P300 response, no signals were identified as reflecting the *learning* contrast (changepoint-oddball) after correcting for multiple comparisons using a permutation test (*Figure 3—figure supplement 2*).

## Behavioral relevance of the P300

Competing theories posit different functional roles for the signal underlying the P300. In particular, some theories suggest that the P300 reflects a general surprise signal, whereas others attribute a more specific role in accumulating information, for example about the current state of the world. To test how the P300 may relate to learning behavior in our task we extracted trial-to-trial measures of these components by taking the dot product of the cluster t-map and each single trial ERP (*Figure 4A*; *Collins and Frank, 2018*). The dot product indexes the degree to which a single trial ERP displays the profile of a given spatiotemporal cluster, thereby allowing us to test the degree to which the measured signal on any given trial might relate to behavior. We then examined how trial-to-trial behavioral updates in shield position related to these single trial EEG signal strengths using a regression model similar to that employed in the behavioral analysis (*Figure 4B*). The regression model included two key terms to characterize the influence of 1) the multiplicative interaction of prediction error with the EEG signal strength, and 2) the interaction between prediction error, EEG signal strength and condition. The first EEG-based term provided a measure of the relationship between learning and the P300 that was independent of condition, and thus allowed us to test the prediction that the P300 reflects a *direct learning* signal (*Figure 4C*, left). The second EEG-based term provided a measure of the relationship between learning and the P300 that depended on condition (*conditional learning*), and thus allowed us to test the prediction that any learning impact of the P300 is bidirectionally sensitive to the source of surprise (*Figure 4C*, right).

Indeed, participant learning behavior systematically related to trial-by-trial measures of the P300, but only in a manner that depended critically on task condition. *Direct learning* coefficients from the model revealed that the P300 signal was not systematically related to learning in the same manner across both conditions (*Figure 4D*, left; mean/SEM = −0.014/0.01, dof = 38, t = −1.5, p=0.14). In contrast, *conditional learning* coefficients tended to be positive (mean/SEM = 0.09/0.02, dof = 38, t = 4.7, p=$3\times10^{-5}$), albeit with considerably heterogeneity across participants (*Figure 4D*, Right). Individual differences in the degree to which the P300 conditionally predicted learning were related to individual differences in the degree to which participant updates were conditionally responsive to surprise, as measured by our behavioral regression model (*Figure 4E*). In particular, the participants who showed the greatest behavioral modulation of learning according to surprise and condition (e. g., the behavioral effect that one would expect to be mediated by a conditional learning signal) tended to also have the highest *conditional learning* coefficients indicating the degree to which P300 conditionally predicted learning (*Figure 4E*; r = 0.54, p=$3\times10^{-4}$). Learning rate predictions derived from the EEG-based regression model show that higher P300 signal strength predicts more

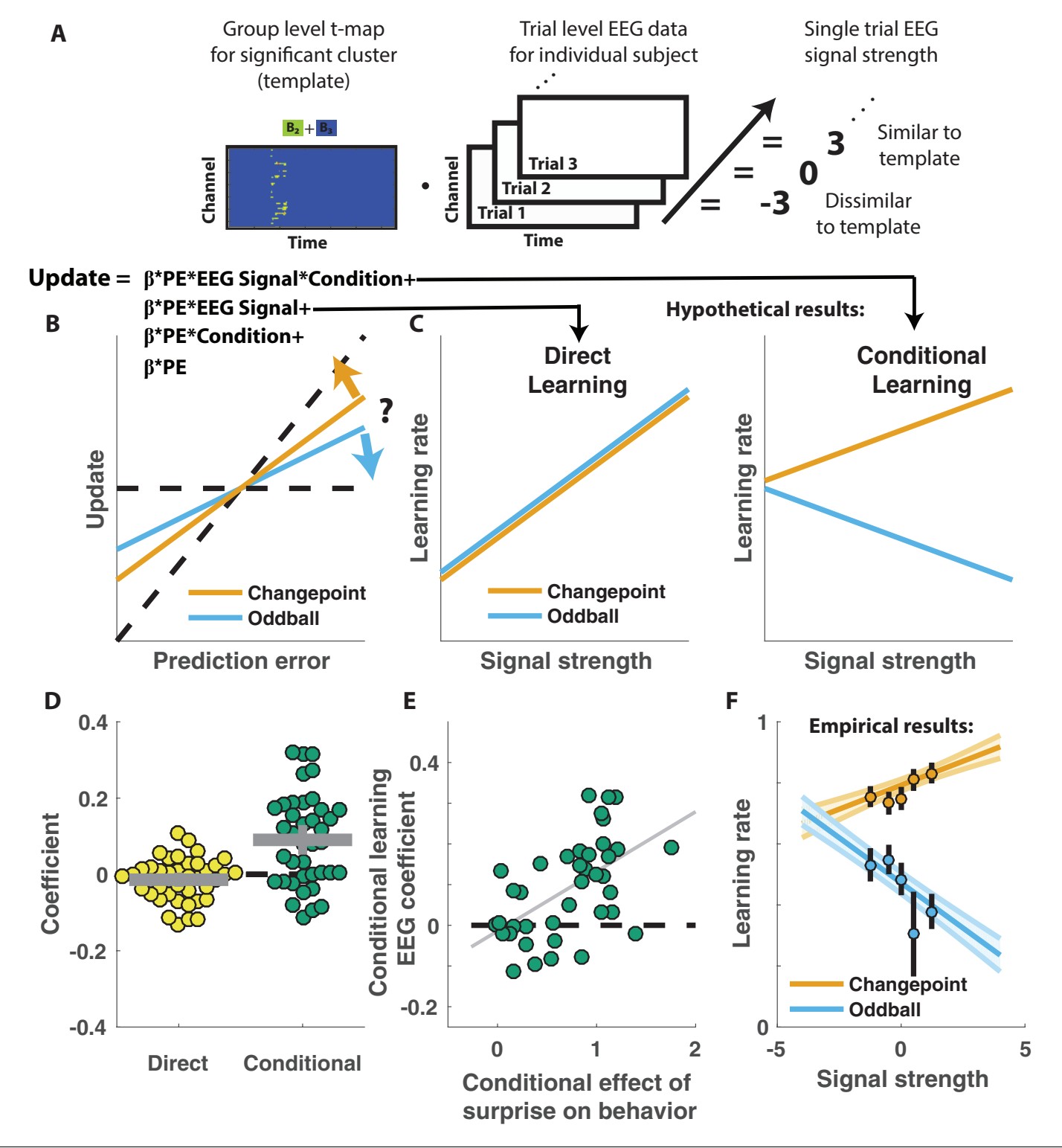

**Figure 4.** Central positivity predicts learning in opposite directions for changepoint and oddball contexts. (**A**) T-maps corresponding to significant spatiotemporal clusters were used as templates to estimate trial-by-trial signal strength. (**B**) Single trial updates for each participant were fit with a regression model that included additional terms to describe 1) the degree to which learning was increased on trials in which the EEG signal was stronger (PE times EEG signal) as would be expected for a canonical learning signal and 2) the degree to which learning was conditionally modulated by the EEG signal (PE times condition times EEG signal) as would be expected for a surprise signal that influenced downstream learning computations. (**C**) Hypothetically, the learning rate (slope of the relationship between updates and prediction errors) might increase for stronger EEG signals (left)

*Figure 4 continued on next page*

*Figure 4 continued*

which would be captured by the PE times EEG direct learning regressor. Alternatively, the learning rate may increase for stronger EEG signals in the changepoint condition and decrease for stronger EEG signals in the oddball condition, as measured by the *conditional learning* regressor. (D) Individual participant coefficients (points) revealed no significant main effect of P300 signals on direct learning (yellow), but a strong positive interaction (*conditional learning*) effect (green), indicating that the signals were differentially predictive of learning in the changepoint and oddball conditions. (E) Individual differences in the degree to which P300 signals conditionally predicted learning (ordinate) were related to differences in the degree to which those participants conditionally modulated updating according to surprise (abscissa, same as orange points in *Figure 2F*). Participants who conditionally adjusted their treatment of surprising information the most (right most points) had P300 signals that had stronger conditional relationships to their updating behavior (upper points). (F) Learning rates predicted by the regression model (ordinate) increased as a function of P300 signal strength (abscissa) in the changepoint condition (orange) but decreased as a function of signal strength in the oddball condition (blue). Regression model predictions line up well with learning rates estimated for five bins of P300 signal strength across the two conditions (points/lines reflect mean/SEM learning rate estimated across participants for each bin).

DOI: https://doi.org/10.7554/eLife.46975.008

The following figure supplement is available for figure 4:

**Figure supplement 1.** Conditional learning effect was driven by positive effects of EEG signals on learning in the changepoint condition and negative effects of EEG signals on learning in the oddball condition.

DOI: https://doi.org/10.7554/eLife.46975.009

learning in the changepoint condition (*Figure 4E*, orange), but less learning in the oddball condition (*Figure 4E*, blue) and this prediction was validated in a follow-up analysis that separately modeled the effect of EEG on learning in the changepoint and oddball conditions (*Figure 4—figure supplement 1*). Thus, there was a systematic relationship between P300 and learning, but that relationship was oppositely modulated by the task condition and hence the inferred source of surprise.

The relationship between the P300 and participant learning behavior persisted even after controlling for all known sources of variability in learning behavior. To establish this, we conducted a similar analysis to that described above to test whether the P300 displayed *direct* or *conditional* learning relationships to behavior, but also: 1) included an additional predictor term that could account for variability in updating captured by the behavioral regression model (*Figure 2E*), and 2) conducted the analysis in sliding windows of time from 300 to 700 ms after outcome presentation (*Figure 5B*). These modifications allowed us to test if and when the P300 could explain variance in updating behavior that was unrelated to observable task features (*Figure 5A*). Consistent with our previous analysis, *conditional learning* coefficients were positive at early time points within the P300 signal window (peak time = 318 ms, peak mean/SEM coefficient = 0.04/0.01) and the extent of contiguous positive coefficients was more extreme than would be expected due to chance (permutation test for cluster mass: mass = 52.8, p=0.01). We also observed in later time points that *direct learning* coefficients tended to be negative across participants, and this negativity was significant after correcting for multiple comparisons (permutation test for cluster mass: mass = 61.8, p=0.01), however this result should be interpreted cautiously given that we did not see a systematic *direct learning* effect before including behavioral predictions in the regression model (*Figure 4D*). Taken together, our results demonstrate that the magnitude of the P300 signal predicted learning increases in changepoint contexts and learning decreases in oddball contexts, and did so beyond what could otherwise be predicted with behavioral modeling alone.

## Discussion

The brain receives a steady stream of sensory inputs, but these inputs differ dramatically from moment to moment in the degree to which they should affect ongoing inferences about the world. People and animals do not treat each datum in this stream as the same, and instead tend to rely more heavily on some pieces of information than others. Identifying the mechanisms through which these adjustments occur could be an important step toward understanding why learning occurs more rapidly in some domains or for some people, yet our understanding of these mechanisms has been heavily conditioned on specific statistical contexts, namely changing environments in which the degree to which one should learn from information is closely coupled to the surprise associated with

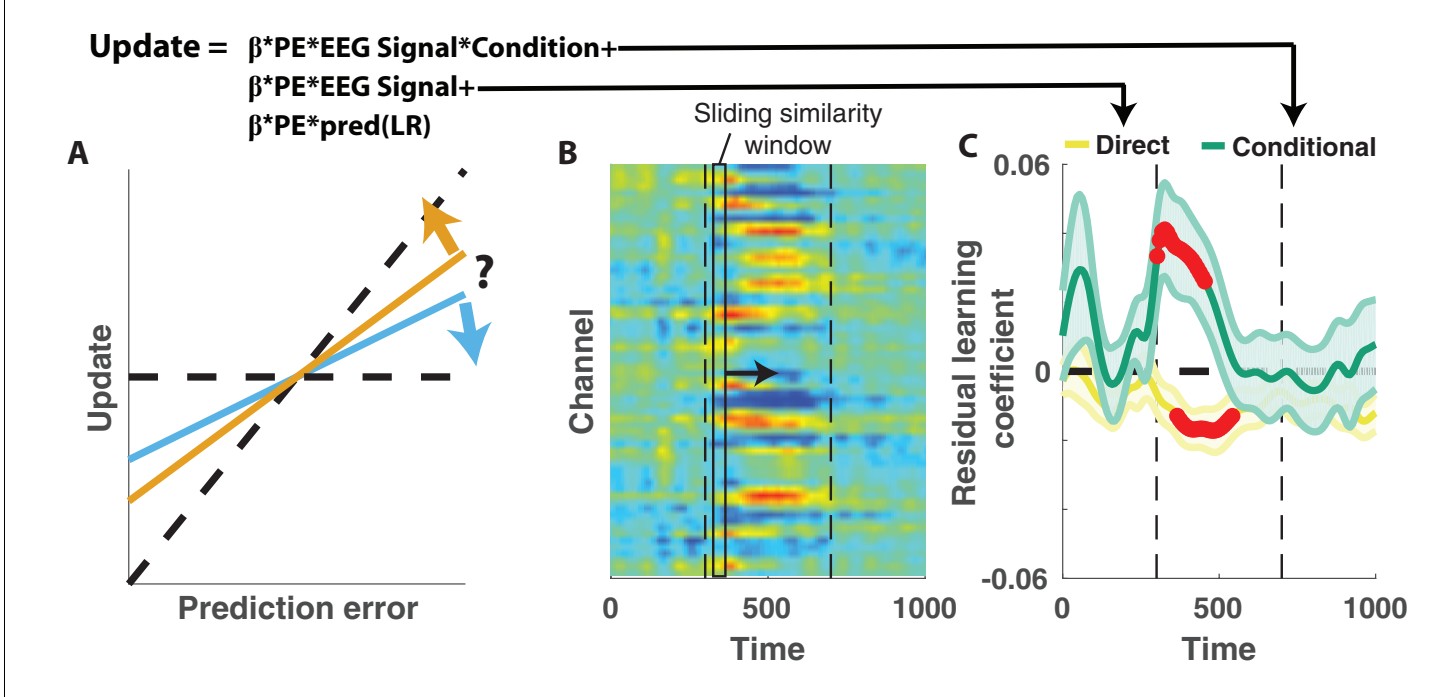

**Figure 5.** Central positivity explains trial-to-trial learning behavior that could not be otherwise captured through behavioral modeling. (A) Single trial updates for each participant were fit with a regression model that included the best estimates of learning rate provided by our behavioral regression model (β times PE times pred(LR)) as well as additional terms to describe the degree to which learning was increased on trials in which an EEG signal was present or the degree to which learning was contextually modulated by the EEG signal. (B) Regression was performed using EEG signal strength computed in 40 ms sliding windows following the time of the outcome. EEG signal strength within a given sliding window for a given trial was computed as the dot product of the baseline corrected ERPs within the window (depicted by rectangle) and the unthresholded t-statistic map for the Changepoint plus Oddball contrast (depicted in heatmap). This measure allowed us to examine how the relationship between P300 and behavior evolved over the time course of the P300 signal [300–700 ms, marked with vertical dotted lines]. (C) *Direct learning* (ordinate; yellow) and *conditional learning* (ordinate; green) coefficients for EEG terms in the regression model are plotted across time (abscissa). Lines/shading reflect mean/SEM coefficients and red points reflect periods over which coefficient values deviated significantly from zero (permutation test on cluster mass: cluster mass/ p value = 61.8/0.01 for *direct learning* and 52.8/0.01 for *conditional learning* coefficients respectively).

DOI: https://doi.org/10.7554/eLife.46975.010

it. Here we examined how relationships between learning and a specific brain signal, the P300 evoked EEG potential, depend on the statistical context that they are measured in.

We show that the P300 relates systematically to learning, but that the direction of this relationship depends critically on the statistical context. In a context where surprising events indicated change-points (*Figure 1C,E*) and participants learned more from surprising information (*Figure 2*), larger P300 responses predicted increased learning (*Figure 4*). In contrast, in a context where surprising events indicated oddballs (*Figure 1B,D*) and participants deemphasized surprising information (*Figure 2*), larger P300 responses predicted reduced learning (*Figure 4*). These context-dependent predictive relationships explained variance in learning beyond what could be captured through computational modeling of behavior alone (*Figure 5*), suggesting that the P300 signal may be involved in adjustments of learning rate, but does so by mediating the subjective response to surprise, rather than translating surprise into a conditionally appropriate learning signal.

## Neural representations of surprise and updating

A key question that has motivated a number of recent studies is how does the brain represent surprise differently than the belief updating it sometimes prescribes. Under most conditions, the degree of surprise is tightly linked to the update that is required. However, recent fMRI studies have exploited cued updating paradigms (*O'Reilly et al., 2013*), irrelevant stimulus dimensions

(*Schwartenbeck et al., 2016*; *Nour et al., 2018*), and complementary statistical contexts (*d'Acremont and Bossaerts, 2016*) in order to tease apart neural representations of surprise and updating. While there are trends that seem to generalize across task boundaries (for example, dorsal anterior cingulate cortex (dACC) reflecting updating in cued updating and irrelevant stimulus dimension paradigms; *O'Reilly et al., 2013*; *Nour et al., 2018*) there is also a good deal of inconsistency across different tasks in terms of the roles of specific signals. For example, even though BOLD responses in dACC were identified as reflecting updating in two studies, they were shown to represent surprise in another (*d'Acremont and Bossaerts, 2016*) and manipulations of statistical context failed to reveal *any* brain regions that provide a pure updating signal (*d'Acremont and Bossaerts, 2016*).

One possible explanation for this discrepancy is that the component processes of updating and non-updating might overlap in some specific paradigms. For example, the oddball outcomes that led to reduced learning in our paradigm and that of d'Acremont and Bossaerts were dissimilar to all previous outcomes and indistinguishable on other feature dimensions (in contrast to *O'Reilly et al., 2013*). Thus, while these outcomes do not contain information pertinent to ongoing beliefs about future outcomes, they did contain information critical for perception, namely that prior expectations should not be used to bias their perceptual representations (*Krishnamurthy et al., 2017*). Interestingly, recent work has suggested that people dynamically adjust the degree to which percepts are biased using systems, including the pupil linked arousal system, that are closely linked to the systems implicated in adjusting learning rate (*Nassar et al., 2012*; *Krishnamurthy et al., 2017*; *Nieuwenhuis et al., 2011*; *Vazey et al., 2018*; *Urai et al., 2017*; *de Gee et al., 2017*). Thus, one possible explanation for the inconsistency in previous studies attempting to dissociate surprise from updating is that these studies have differed in the degree to which they inadvertently manipulated systems for controlling perceptual biases.

Like in the previous fMRI study relying on statistical context to dissociate learning from surprise (*d'Acremont and Bossaerts, 2016*), our EEG results revealed a large number of signals related to surprise and no signals that convincingly reflected learning rate in a context independent manner. This comes as somewhat of a surprise given previous work identifying EEG signals analogous to a late P300 component reflecting surprise, predicting learning and influence on choice even in paradigms where this influence was unrelated to surprise (*Cheadle et al., 2014*; *Jepma et al., 2016*; *Jepma et al., 2018*; *Fischer and Ullsperger, 2013*; *Wyart et al., 2012*). In line with previous work from fMRI studies, we interpret the differences in our results from what might have been predicted based on previous work as pertaining to unique strategy we employed for dissociating learning from surprise through the use of different statistical contexts.

## Mechanisms of learning rate adjustment

Our results, particularly when taken in the context of previous studies examining how the brain adjusts learning in accordance with surprise, constrain possible models of learning rate adjustment in the brain. We show that that the updating P300 signal, which positively predicts learning in changing environments (*Figure 4E*), also negatively predicts learning in a context with infrequent statistical outliers (*Figure 4E*). Thus, in a most basic sense, our results suggest that the P300 signals reflects an early contribution to learning rate adjustment, and that this signal is untangled according to statistical context at some downstream stage of processing. The lack of robust ERP correlates of direct learning signals (*Figure 3—figure supplement 2*) suggests that this downstream process does not have a task-locked electrophysiological signature.

One potential mechanism for learning rate adjustment that fits well with these constraints is the notion that adjustments in learning might be implemented through flexible replacement of state representations (*Collins and Reasoning, 2012*; *Collins and Frank, 2013*; *Wilson et al., 2014*). Learning rate adjustment is adaptive in changing environments because it can effectively partition data relevant to the current predictive context from data that are no longer relevant to prediction (*Adams and MacKay, 2007*; *Wilson et al., 2010*). One possible implementation of this partitioning would be to change the active state representations that serve as the substrate for contextual associations. Recent work has identified signals in OFC, a region implicated in representations of latent states (*Schuck et al., 2016*), that change more rapidly during periods of rapid learning (*Nassar et al., 2019*). If this is indeed the implementation through which learning rate adjustments

occur, observed learning rate signals might actually signal the need to adjust the representation of the latent state.

Interestingly, replacement of the active latent state, or partitioning of data more generally, might also be an effective way to implement the decreased learning observed in response to surprising observations in the oddball condition of our task. In the case of an oddball, one strategy would be to recognize the oddball as having been generated by an alternative causal process (e.g., oddball distribution) and to attribute learning to a latent representation of this process (*Gershman and Niv, 2010*). Under such conditions, implementation would require a surprise signal that reflects the relevance of this oddball latent state. After the new observation is attributed to the oddball context, the system would require a transition back into the original 'non-oddball' state in order to make a prediction that is unaffected by the most recent oddball outcome. The more effectively surprise is recognized and responded to through latent state changes (e.g., the stronger the surprise signal) the more effectively this implementation would partition an oddball observation from ongoing beliefs about the standard generative process, and therefore the smaller learning rates would be. Thus, one mechanistic interpretation of the P300 results might be that it is providing a partitioning signal that results in transitions in the internal latent state representation, which can either increase or decrease learning depending on the statistical context.

## Implications for theories of P300 function

We took a data driven approach to identifying signals that related to surprising outcomes in different statistical contexts. The primary signal that we identified, however, was similar in timing (*Figure 3D and F*), location (*Figure 3B*), and sensitivity to surprise (*Figure 3E and G*) to those previously reported for the P300 (*Kopp et al., 2016*; *Kolossa et al., 2012*). The topography of our spatiotemporal cluster changed over time from frontocentral to centroparietal, consistent with inclusion of both an early frontocentral P3a component as well as a later centroparietal P3b component. Thus, although our methods were agnostic to detection of a specific signal, we interpret our results in the context of the larger literature relating to P300 signaling.

Our findings are consistent with a number of studies that have demonstrated the P300 is related to surprise (*Jepma et al., 2016*; *Donchin, 1981*; *Wessel, 2018*; *Garrido et al., 2016*), but extend them to reveal how the P300 differentially relates to learning in different contexts. Our results are inconsistent with standard interpretations of the context updating theory of the P300 in which context is defined as a working memory for an observable stimulus (*Donchin, 1981*; *Donchin and Coles, 1988*; *Polich, 2003*; *Polich, 2007*), as under this definition a larger P300 should always lead to more learning. However, if the updated 'contexts' were defined in terms of the latent states described above, the predictions of the context updating theory would indeed match our results. Thus, our results can constrain potential interpretations of the context updating theory, although they do not falsify the theory altogether. Nor do our results directly conflict with other prominent theories of P300 signaling including the idea that central parietal positivity might reflect accumulated evidence for a particular decision or course of action (*Kelly and O'Connell, 2013*; *O'Connell et al., 2012*), or anticipate the need to inhibit responding (*Wessel, 2018*; *Wessel and Aron, 2017*), as both of these theories could be framed in terms of the latent states above (e.g.. the accumulated evidence for a change in latent state or the need to inhibit responding until the appropriate latent state is loaded). Thus, our results do not arbitrate between these theories, but do require expansion of their interpretation (to include latent variables involved in the generation of outcomes) and also highlight their implications for learning when mechanistic interpretations are refined and applied to our task and data.

Confirming our proposed mechanistic interpretation of these results in terms of latent states would require future studies more closely relating P300 signals to purported state representations (*Nassar et al., 2019*). Furthermore, given that our study relied completely on computational modeling and correlations with behavior, our results raise important questions as to whether the observed associations could be manipulated directly pharmacologically or through biofeedback paradigms. Thus, our work provides new insight into the underlying mechanisms of learning rate adjustment and the role of the P300 in this process, but leaves many unanswered questions to be addressed in future research.

## Materials and methods

### Participants

Participants were recruited from the Brown University community: n = 39, 22 female, mean age = 20.2 (SD = 3.1, range = 18–36). Data from all 39 participants was included for both behavioral and EEG analysis. Sample size was selected based on a recent EEG study using a similar task and focusing on the P300 (*Jepma et al., 2016*). All human subject procedures were approved by the Brown University Institutional Review Board and conducted in agreement with the Declaration of Helsinki.

### Cannon task

Participants performed a modified predictive inference task that is available on GitHub (*Bruckner, 2019*; copy archived at https://github.com/elifesciences-publications/AdaptiveLearning) and was programmed in Matlab (Mathworks, Natick, MA, USA), using the Psychtoolbox-2 (http://psychtoolbox.org/) package. The task was based on predictive inference tasks in which participants are asked to predict the next in a series of outcomes (*Nassar et al., 2010*; *Nassar et al., 2012*; *Nassar et al., 2016*), but differed from previous such tasks the following ways: (1) the outcomes were generated from both changepoint and oddball processes to dissociate learning from surprise, (2) information necessary for performance evaluation was not available at time of outcome so that signals related to belief updating could be dissociated from valenced performance evaluation signals, (3) the task space was circular, and (4) the generative process was cast in terms of a cannon shooting cannonballs.

Participants were instructed to place a shield at some position along a circle subtending 5 degrees of visual angle in order to maximize the chances of catching a cannonball that would be shot on that trial (*Figure 1A*). During an instructional training period, the generative process that gave rise to cannonball locations was made explicit to participants. During this phase, participants were shown a cannon in the center of the screen. On each trial, a cannonball would be 'shot' from that cannon with some angular variability (Von Mises distributed 'Noise', concentration = 10 degrees). A key manipulation in our design was how the aim of the cannon evolved from one trial to the next. The cannon would either (1) remain stationary on the majority of trials and re-aim to a random angle with an average hazard rate of 0.14 (changepoint condition) or (2) change position slightly from one trial to the next according to a Von Mises distributed random walk with mean zero and concentration 30 degrees (oddball condition). In the changepoint condition, all cannonballs were displayed as originating at the cannon in the center of the circle, whereas in the oddball condition a small fraction (0.14) of trials were oddballs, in which the cannonball location was sampled uniformly across the entire circle and the cannonball appeared without a trajectory.

Each experimental condition was preceded by (1) instructions that included an explicit description of the generative process for that condition and (2) a set of training trials with the cannon visible such that the generative process could be observed directly. These instructions and training trials were designed to ensure that all participants were aware of the statistical context, so as to improve our ability to detect EEG signals that related to it.

After completing the instructional training, in which the generative process was fully observable, participants were asked to perform the same basic task without being able to see the cannon. In this experimental phase participants were forced to use knowledge of the generative structure gained during training, along with the sequence of prior cannonball locations, in order to infer the aim of the cannon and to inform shield placement. When making a prediction, tick marks on the circle indicated the locations of the cannonball and shield placement from the previous trial. Participants completed four blocks of 60 trials for each task condition (changepoint and oddball) in order randomized across participants. The 240 experimental trials for each condition always followed the instructional training period for that condition in order to minimize ambiguity over which generative structure was giving rise to the experimental outcomes.

On each trial of the experimental task, participants would adjust the position of the shield through key presses (starting at the shield position from the previous trial) until they were satisfied with its location (*Figure 1A*; prediction phase). After participants locked in their prediction (through a key press) there was a 500 ms delay and then the cannonball location was revealed for 500 ms

(*Figure 1A*; outcome phase). The cannonball then disappeared for 1000 ms before it reappeared, along with a full depiction of the participants shield (*Figure 1A*; shield phase). The shield was always centered on the position indicated by the participant during the prediction phase, but differed in size from one trial to the next in a random and unpredictable fashion that ensured subjects could not predict whether they would successfully 'catch' the cannonball during the outcome phase. Thus, information provided during the outcome phase provided all necessary information to update beliefs about the cannon aim, but did not contain sufficient information to determine whether the cannonball would be successfully blocked on the trial. In addition to trial feedback provided during the shield phase, participants were also provided information about their performance at the end of each block that included the fraction of cannonballs that were blocked. Participants were paid an incentive bonus at task completion that was based on the number of cannonballs that were blocked.

## Computational model

Optimal inference in the changepoint condition would require considering all possible durations of stable cannon position (*Adams and MacKay, 2007*; *Wilson et al., 2010*) but can be approximated by collapsing the mixture of predictive distributions expected to arise from this optimal solution into a single Gaussian distribution, which approximates the posterior probability distribution over cannon locations, achieves near optimal inference, reduces to an error driven learning rule in which learning rate is adjusted from moment to moment according to environmental statistics, and provides a detailed account of human behavior (*Nassar et al., 2010*; *Nassar et al., 2016*). Similarly, the ideal observer for the oddball generative process would require tracking the predictive distributions and posterior probabilities associated with each possible sequence of oddball/non oddball trials that could have preceded the time step of interest. Like in the changepoint condition, this algorithm can be simplified by approximating the set of all possible predictive distributions with a single Gaussian distribution, leading to an error driven learning rule in which learning rate is adjusted dynamically from trial to trial, allowing us to derive normative prescriptions for learning for both conditions (see supplementary material for full derivation).

While the normative model for the changepoint condition has been described elsewhere (*Nassar et al., 2016*) the analogous model for the oddball condition is not, and thus we describe the normative account of oddball learning in full detail. In order to minimize the differences between experienced and modeled latent variables, we formulated our model in terms of the prediction errors made by participants on each trial (rather than those that would have been made by the model) (*Nassar et al., 2016*). On each trial of the oddball condition, the normative model: (1) updated its representation of uncertainty, (2) observed a prediction error and computed the probability that the prediction error reflects an oddball, (3) computed the normative learning rate by combining uncertainty (step 1) and oddball probability (step 2), (4) adjusted prediction about cannon position according learning rate and prediction error.

Relative uncertainty, which reflects the fraction of uncertainty about an upcoming cannonball location that is due to imperfect knowledge of the cannon aim and is analogous to the Kalman gain, was updated on each trial according to the most recent observation (which should decrease uncertainty about cannon position) and the expected drift in the aim of the cannon occurring between trials (which should increase uncertainty about cannon position). Given that relative uncertainty is expressed as a fraction of total uncertainty, it is useful to think of the numerator of the fraction, or the estimation uncertainty over possible cannon aims, which is the variance on a gaussian mixture distribution and is updated as follows:

$$\sigma_\mu^2 = \Omega_t \frac{\sigma_N^2 \tau_t}{1 - \tau_t} + (1 - \Omega_t)\sigma_N^2 \tau_t + \Omega_t(1 - \Omega_t)(\delta_t \tau_t)^2 + \sigma_{drift}^2$$

where $\Omega_t$ is the probability that an oddball occurred on trial $t$, $\sigma_N^2$ reflects the variance on the distribution of cannonball locations around the true cannon aim (noise), $\tau_t$ reflects the relative uncertainty on trial $t$, $\delta_t$ is the prediction error made in predicting the outcome on trial $t$, and $\sigma_{drift}^2$ reflects the degree to which the cannon position drifts from one trial to the next. The first two terms in the model reflect the oddball and non-oddball contributions to the updated uncertainty, the third term reflects uncertainty resulting from the difference between predictions for trial *t+1* conditioned on an oddball or non-oddball having occurred on trial $t$, and the last term reflects uncertainty resulting

from the expected drift of the cannon position between trials. Relative uncertainty for trial $t+1$ is then updated as the updated fraction of uncertainty about the upcoming outcome that is attributable to imprecise knowledge of the true cannon position, rather than to noise in the distribution of exact cannonballs around that position:

$$\tau_{t+1} = \frac{\sigma_\mu^2}{\sigma_\mu^2 + \sigma_N^2}$$

The updated relative uncertainty, along with assumed knowledge of the overall noise and hazard rate, were used to calibrate the oddball probability associated with each new prediction error:

$$\Omega_{t+1} = \frac{\frac{H}{2\pi}}{\frac{H}{2\pi} + (1-H)\,\Re\left(\delta_{t+1}; 0, \frac{\sigma_N^2}{1-\tau_{t+1}}\right)}$$

Where H is the average hazard of an oddball (0.14) and $\delta_{t+1}$ is the new prediction error, and the second term in the denominator reflects the probability density on a normal distribution centered on the predicted location and with variance derived from relative uncertainty. The model's prediction about cannon aim was then updated according to a fraction of the prediction error $\delta_{t+1}$ with the exact fraction, or learning rate, determined according to the updated uncertainty and oddball probability:

$$\alpha_{t+1} = \tau_{t+1} - \Omega_{t+1}\tau_{t+1}$$

Note that relative uncertainty ($\tau_{t+1}$) contributes positively to the learning rate, whereas oddball probability ($\Omega_{t+1}$) reduces the learning that would otherwise be dictated by the current level of uncertainty.

## Behavioral analysis

Two key behavioral measures were extracted from each trial. First, the *prediction error* on a trial was defined as the circular distance between the cannonball location and the shield position for that trial. Second, the *update* on a given trial was defined as the circular distance between the shield position on that trial and the shield position on the subsequent trial (e.g., the updated shield position). In order to better understand the computational factors governing adjustments in shield position, we fit *updates* with a linear model that included an intercept term to model overall biases in learning along with a prediction error term to capture general tendencies to adjust the shield towards the most recent cannonball location. The model also included additional terms to model how the influence of recent cannonball locations changed dynamically according to task context. These terms included: (1) prediction error times uncertainty interaction (to model how much more participants updated shield position under conditions of uncertainty – as assessed by the computational model), (2) prediction error times surprise (where surprise was indexed by changepoint probability or oddball probability from computational model depending on the context), (3) prediction error times surprise times condition (where condition was +1 for changepoint blocks and −1 for oddball conditions), (4) prediction error times block (a categorical variable indicating whether the shield 'blocked' the most recent cannonball. The model was fit to updates from each participant individually, excluding updates in response to outcomes immediately following oddballs, as such updates are difficult to attribute to learning (e.g., movements toward recent outcome) rather than memory (returning to a pre-oddball location). Nonetheless, inclusion of these trials in our behavioral models does not change the primary results reported here.

Unlike standard regression in which the error distribution is assumed to be normal, our model imposed a circular (Von-Mises) distribution of errors around the predicted update. Maximum posterior coefficients for each individual subject were estimated using the fmincon optimization tool in Matlab (Mathworks, Natick, MA, USA) and t-tests were performed on the regression coefficients across participants to test for significant contributions of each term to update behavior. Weak Gaussian zero-centered priors were included to regularize coefficients of interest. In the purely behavioral analysis the width of priors over coefficients on standardized predictors was set to 5. In the analyses that included EEG predictors, EEG-based predictors were regularized using a zero centered Gaussian prior with a standard deviation of 0.1 (as compared to a standard deviation of 1 for

the predicted learning rates from the behavioral model) making the regularization for the EEG terms stronger than that on the competing behavioral prediction term by a factor of 10 (thereby allowing preferential explanation of shared variance by the other terms in the model).

## EEG acquisition

EEG was recorded from a 64-channel Synamps2 system (0.1–100 Hz bandpass; 500 Hz sampling rate). Data were collected using CPz as a reference channel and re-referenced to the grand mean for analysis. Continuous EEG data was epoched with respect to the outcome presentation for each trial. Preprocessing was done manually in Matlab (Mathworks, Natick MA) using the EEGLAB toolbox (https://sccn.ucsd.edu/eeglab/index.php) as described previously (*Collins and Frank, 2018*) and included the following steps: (1) epoching and alignment to outcome onset, (2) epoch rejection by inspection, (3) channel removal and interpolation by inspection, (4) bandpass filtering [0.05–15 hz], (5) removal of blink and eye movement components using ICA.

## EEG analysis

EEG Data for individual participants were analyzed using a mass univariate approach. Specifically, the trial series EEG data for a given participant, channel, and time relative to outcome onset was regressed onto an explanatory matrix that included the following explanatory variables: (1) intercept, (2) changepoint, (3) oddball, (4) condition, (5) block. Explanatory variables 2 and three were binary variables marking trials in which a surprising event occurred (i.e. changepoint or oddball) whereas four reflected the overall task context (i.e. whether oddballs or changepoints were present in the current statistical context), and five conveyed whether the participant successfully 'blocked' the cannonball on each trial. Surprise and learning contrasts were created as the sum and difference of the changepoint and oddball coefficients, respectively. T-statistics were computed across subjects to assess the consistency of contrasts at each electrode and timepoint.

T-statistic maps were thresholded (cluster forming threshold of p=0.001, two tailed) and spatio-temporal clusters were identified as temporally and/or spatially contiguous signals that shared a common sign of effect and exceeded the cluster-forming threshold. Cluster mass was computed as the average absolute t-statistic within a cluster times its size (number of electrode timepoints contained within it). Cluster mass for each spatiotemporal cluster was compared to a permutation distribution for cluster mass generated using sign flipping to correct for multiple comparisons (*Nichols and Holmes, 2002*).

Trial-to-trial EEG analyses were conducted by computing the dot product of the t-statistic map for a given spatiotemporal cluster and the ERP measured on a given trial. The resulting measure of EEG signal strength was then z-scored across all trials and included in a behavioral regression model to explain trial-to-trial updating behavior. Like for the behavioral analyses, trial-to-trial updates were regressed onto an explanatory matrix that included intercept and prediction error terms to capture updating biases and static tendencies to update toward recent cannonball locations. In addition, EEG informed linear model included (1) the interaction between the EEG signal strength computed above and prediction error (*direct learning*), and (2) the three-way interaction between EEG signal strength, prediction error, and condition (*conditional learning*). Positive *direct learning* coefficients indicated an unconditional increase in learning for trials in which EEG signal strength was greater, whereas positive *conditional learning* coefficients indicated a positive relationship between EEG signal strength and learning in the changepoint condition but a negative relationship between EEG signal strength in the oddball condition.

In order to test if and when the EEG-updating relationships could explain behavior beyond the best descriptions afforded by our behavioral model, we applied the method described above with two changes. First, we computed EEG signal strength in sliding windows of time by masking the unthresholded *surprise* t-statistic map with a sliding 40 ms window and taking the dot product of the masked t-map and masked ERPs from each trial. We included signal strength calculated in this way in a regression to explain updating behavior as described above, except that we also included the predicted update from our purely behavioral model, as a competing explanatory variable. *Direct learning* and *conditional learning* coefficients for each participant were smoothed in time by convolving them with a Gaussian kernel (std = 8 ms). T-statistics were computed for each sliding window and coefficient across participants and temporal clusters of extreme t-statistics were formed using a

cluster-forming threshold of p<0.05. Cluster mass was computed as described above and compared to permutation distribution created with iterative sign-flipping (10000 permutations) to estimate a cluster-corrected p-value.

## Acknowledgements

We would like to thank Julie Helmers and Andrea Mueller for their help collecting EEG and behavioral data Rob Chambliss for help cleaning EEG data, and Romy Frömer and Rachel Ratz-Lubashevsky for helpful discussion. This work was funded by NIH grants F32MH102009 and K99AG054732 (MRN), NIMH R01 MH080066-01 and NSF Proposal #1460604 (MJF). RB was supported by a Promos travel grant from the German Academic Exchange Service (DAAD). The funders had no role in study design, data collection and analysis, decision to publish or preparation of the manuscript.

## Additional information

### Competing interests

Michael J Frank: Senior editor, *eLife*. The other authors declare that no competing interests exist.

### Funding

| Funder | Grant reference number | Author |
| --- | --- | --- |
| National Institute of Mental Health | F32MH102009 | Matthew R Nassar |
| National Institute on Aging | K99AG054732 | Matthew R Nassar |
| National Institute of Mental Health | R01 MH080066-01 | Michael J Frank |
| National Science Foundation | 1460604 | Michael J Frank |
| German Academic Exchange Service London | Promos travel grant | Rasmus Bruckner |

The funders had no role in study design, data collection and interpretation, or the decision to submit the work for publication.

### Author contributions

Matthew R Nassar, Conceptualization, Resources, Formal analysis, Funding acquisition, Investigation, Methodology, Writing—original draft, Writing—review and editing; Rasmus Bruckner, Conceptualization, Resources, Software, Methodology, Writing—review and editing; Michael J Frank, Conceptualization, Writing—review and editing

### Author ORCIDs

Matthew R Nassar (iD) https://orcid.org/0000-0002-5397-535X
Rasmus Bruckner (iD) http://orcid.org/0000-0002-3033-6299
Michael J Frank (iD) http://orcid.org/0000-0001-8451-0523

### Ethics

Human subjects: Informed consent was obtained from each participant in the study and all procedures were performed in accordance with the Declaration of Helsinki. All procedures were approved by the Brown University Institutional Review Board (Brown University Federal Wide Assurance #00004460).

### Decision letter and Author response

Decision letter https://doi.org/10.7554/eLife.46975.019
Author response https://doi.org/10.7554/eLife.46975.020

## Additional files

### Supplementary files

• Transparent reporting form
DOI: https://doi.org/10.7554/eLife.46975.011

### Data availability

All analysis code has been made available on GitHub (https://github.com/learning-memory-and-decision-lab/NassarBrucknerFrank_eLife_2019.git; copy archived at https://github.com/elifesciences-publications/NassarBrucknerFrank_eLife_2019). All behavioral and EEG data has been made available on Dryad (https://doi.org/10.5061/dryad.570pf8n).

The following dataset was generated:

| Author(s) | Year | Dataset title | Dataset URL | Database and Identifier |
|---|---|---|---|---|
| Matthew R Nassar, Rasmus Bruckner, Michael J Frank | 2019 | Statistical context dictates the relationship between feedback-related EEG signals and learning | https://doi.org/10.5061/dryad.570pf8n | Dryad Digital Repository, 10.5061/dryad.570pf8n |

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

## Appendix 1

DOI: https://doi.org/10.7554/eLife.46975.012

## Computational modeling

*Changepoint* condition generative process

*Oddball* condition generative process

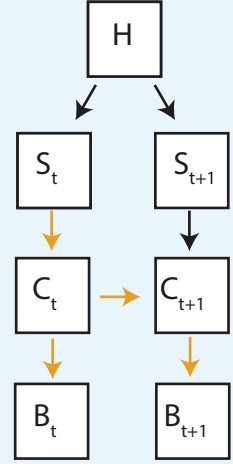
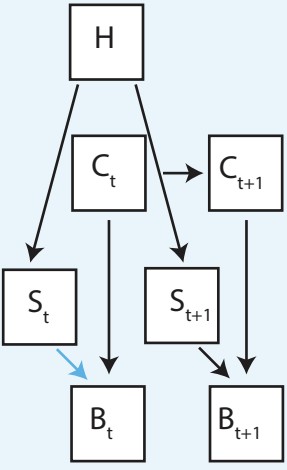

**Legend:**

H = Hazard rate (Frequency of S=1)
S = binary changepoint/oddball variable
C = Mean of generative distribution (cannon aim)
B = Cannball location

**Appendix 1—figure 1.** Graphical generative model for changepoint (left) and oddball (right) task conditions. Subscripts denote time, colored arrows depict the causal influence of an unlikely event (oddball or changepoint) on current and future outcomes. Note that changepoints (left) affect both current and future outcomes (yellow arrows) whereas oddballs (right) only affect current outcomes (blue arrow).
DOI: https://doi.org/10.7554/eLife.46975.013

### Derivation of normative learning model for changepoint condition

The generative process in the changepoint condition can be defined in terms of the following sampling statements:

$$H = 0.125$$

$$S_t \sim Bernoulli(H)$$

$$C_t \mid S_t = 1 \sim Uniform(0, 359)$$

$$C_t \mid S_t = 0 \sim Dirac\ Delta(C_{t-1})$$

$$B_t \sim Von\ Mises\ (C_t\,,\ \sigma)$$

Here we assume that the hazard rate is known, given the extensive training participants receive under visible cannon conditions, however the more general case where hazard rate is unknown is similar and has been addressed elsewhere (**Wilson et al., 2010**). Note that the generative hazard rate differs from the empirical hazard rate (0.14) because the first trial of each block was considered to be a changepoint.

The inference problem is thus to infer the current cannon aim based on the sequence of observed cannonballs, which has a recursive solution according to the conditional probability:

$$p(C_t|B_{1:t}) = \frac{p(C_t, B_{1:t})}{p(B_{1:t})} \tag{1}$$

that can be derived based on the above-defined random variables. The joint distribution in the numerator can be expanded and partitioned according to the generative graph as follows:

$$
\begin{aligned}
p(C_t, B_{1:t}) \ &= \sum_{S_t} \int_{C_{t-1}} p(B_{1:t}, C_t, C_{t-1}, S_t) \\
&= \sum_{S_t} \int_{C_{t-1}} p(B_t|B_{1:t-1}, C_t, C_{t-1}, S_t) p(B_{1:t-1}, C_t, C_{t-1}, S_t) \\
&= \sum_{S_t} \int_{C_{t-1}} p(B_t|C_t) p(B_{1:t-1}, C_t, C_{t-1}, S_t) \\
&= \sum_{S_t} \int_{C_{t-1}} p(B_t|C_t) p(C_t|B_{1:t-1}, C_{t-1}, S_t) p(B_{1:t-1}, C_{t-1}, S_t) \\
&= \sum_{S_t} \int_{C_{t-1}} p(B_t|C_t) p(C_t|C_{t-1}, S_t) p(B_{1:t-1}, C_{t-1}, S_t) \\
&= \sum_{S_t} \int_{C_{t-1}} p(B_t|C_t) p(C_t|C_{t-1}, S_t) p(C_{t-1}|B_{1:t-1}, S_t) p(B_{1:t-1}, S_t) \\
&= \sum_{S_t} \int_{C_{t-1}} p(B_t|C_t) p(C_t|C_{t-1}, S_t) p(C_{t-1}|B_{1:t-1}) p(B_{1:t-1}, S_t) \\
&= \sum_{S_t} \int_{C_{t-1}} p(B_t|C_t) p(C_t|C_{t-1}, S_t) p(C_{t-1}|B_{1:t-1}) p(B_{1:t-1}|S_t) p(S_t) \\
&= \sum_{S_t} \int_{C_{t-1}} p(B_t|C_t) p(C_t|C_{t-1}, S_t) p(C_{t-1}|B_{1:t-1}) p(S_t) p(B_{1:t-1})
\end{aligned}
\tag{2}
$$

and the marginal distribution in the denominator is obtained from

$$p(B_{1:t}) = p(B_t|B_{1:t-1}) p(B_{1:t-1}) = \int_{C_t} p(C_t, B_{1:t}) \tag{3}$$

which together thus yields

$$p(C_t|B_{1:t}) = \frac{\sum_{S_t} \int_{C_{t-1}} p(B_t|\,C_t)\, p(C_t|\,C_{t-1}, S_t) p(C_{t-1}|\,B_{1:t-1}) p(S_t)}{p(B_t\,|\,B_{1:t-1})} \tag{4}$$

Thus, on each trial it is possible to infer the probability distribution over possible cannon locations given all previous cannonball locations using a recursive rule that updates a prior inference from the previous trial $p(C_{t-1}|B_{1:t-1})$ according to the appropriate transition function $p(C_t\,|S_t,\,C_{t-1})$ and the likelihood of the observed cannonball location from this trial $p(B_t\,|\,C_t)$. Expanding the summation over the changepoint variable reveals that $p(C_t|B_{1:t})$ is a mixture with two components:

$$p(C_t|B_{1:t}) = \frac{\int_{C_{t-1}} p(B_t|C_t)p(C_t|S_t=1, C_{t-1})Hp(C_{t-1}|B_{1:t-1})+}{\int_{C_{t-1}} p(B_t|C_t)p(C_t|S_t=0, C_{t-1})(1-H)p(C_{t-1}|B_{1:t-1})} \Big/ p(B_t|B_{1:t-1})$$

(5)

Where the first component reflects the 'changepoint' predictive distribution and the latter reflects the 'non changepoint' predictive distribution. In principle, we could maintain this mixture distribution and add a new mixture component with each observation (*Appendix 1—figure 2*, top), producing exact parametric inference at a computational cost that scales linearly with time (*Adams and MacKay, 2007*). However, instead, we approximate the mixture distribution by replacing it with a Gaussian distribution that shares the same mean and variance as the full mixture (*Appendix 1—figure 2*, bottom). The mean of the updated mixture distribution can be approximated with the weighted sum of the two mixture components:

$$\hat{c}_{t+1} = (\hat{c}_{t+1}|s_t=1)\Omega_t + (\hat{c}_{t+1}|s_t=0)(1-\Omega_t)$$

(6a)

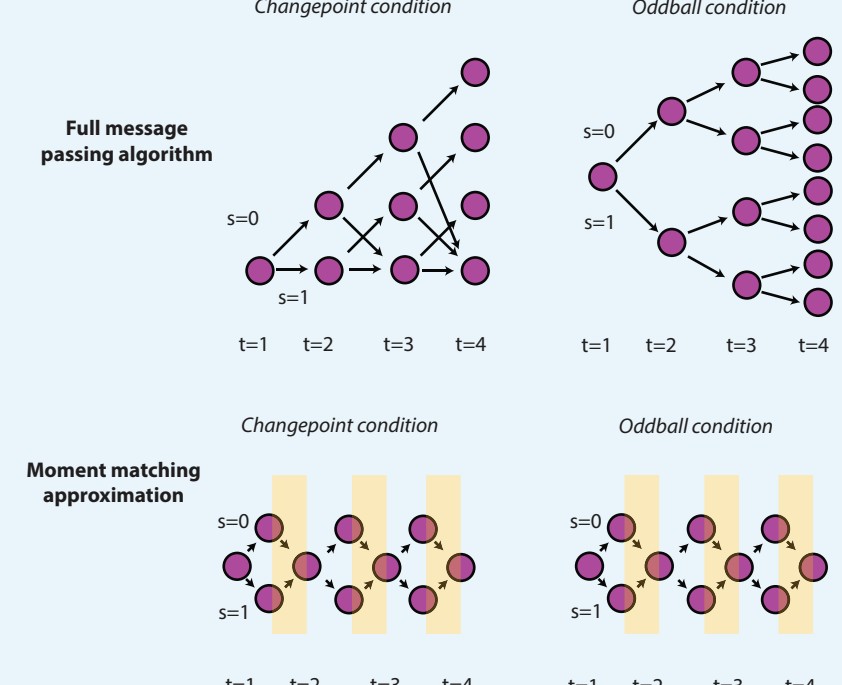

**Appendix 1—figure 2.** Optimal and approximate inference through message passing algorithms. Exact parametric solutions to inference in the changepoint (left) and oddball (right) are possible for a given event history (e.g., S is known at all timesteps). Exact solutions can be approximated by adding an additional step to the full message passing algorithm, in which the mixture distribution (composed of the S = 0 and S = 1 node) is approximated with a Gaussian distribution (yellow stripes) by matching the first two moments of the distributions (*Nassar et al., 2012*).

DOI: https://doi.org/10.7554/eLife.46975.014

In the case of a changepoint, the mean of the predictive distribution over cannon aim $(\hat{c}_{t+1}|s_t=1)$ is equivalent to the most recent outcome $B_t$. The mean of the non-changepoint distribution is a convex combination of the prior predictive mean $\hat{c}_t$ and the most recent outcome $B_t$. Together, this yields an error-driven learning rule, in which the influence of unpredicted outcomes, or learning rate, is adjusted on each trial (*Nassar et al., 2010*):

$$
\begin{aligned}
\hat{c}_{t+1} &= (\hat{c}_{t+1}|s_t=1)\Omega_t + (\hat{c}_{t+1}|s_t=0)(1-\Omega_t) \\
&= B_t\Omega_t + ((1-\tau_t)\hat{c}_t + \tau_t B_t)(1-\Omega_t) \\
&= B_t\Omega_t + (\hat{c}_t + \tau_t B_t - \tau_t\hat{c}_t)(1-\Omega_t) \\
&= B_t\Omega_t + (\hat{c}_t + \tau_t(B_t - \hat{c}_t))(1-\Omega_t) \\
&= (\hat{c}_t + (B_t - \hat{c}_t))\Omega_t + (\hat{c}_t + \tau_t(B_t - \hat{c}_t))(1-\Omega_t) \\
&= (\hat{c}_t + \delta_t)\Omega_t + (\hat{c}_t + \tau_t\delta_t)(1-\Omega_t) \\
&= \hat{c}_t\Omega_t + \delta_t\Omega_t + (\hat{c}_t + \tau_t\delta_t)(1-\Omega_t) \\
&= \hat{c}_t\Omega_t + \delta_t\Omega_t + \hat{c}_t - \hat{c}_t\Omega_t + \tau_t\delta_t - \tau_t\delta_t\Omega_t \\
&= \hat{c}_t + \delta_t\Omega_t + \hat{c}_t\Omega_t - \hat{c}_t\Omega_t + \tau_t\delta_t - \tau_t\delta_t\Omega_t \\
&= \hat{c}_t + \delta_t\Omega_t + \tau_t\delta_t - \tau_t\delta_t\Omega_t \\
&= \hat{c}_t + (\tau_t + \Omega_t - \tau_t\Omega_t)\delta_t \\
&= \hat{c}_t + \alpha_t\delta_t
\end{aligned}
\tag{6b}
$$

where $\hat{c}$ is the mean of the predictive distribution over cannon locations, $\delta_t := (B_t - \hat{c}_t)$ is the prediction error, and $\alpha_t := \tau_t + \Omega_t - \tau_t\Omega_t$ is a learning rate that depends on changepoint probability ($\Omega_t$, the integral over $C_t$ in the first term of **Equation 5**) and relative uncertainty $\tau_t$. The contribution of $\Omega_t$ to the learning rate emerges because, as shown above, the mean is equal to the probability-weighted average of the means of the individual mixture components (changepoint and non-changepoint terms in **Equation 6a**) and thus $\Omega_t$ controls the weights of these two terms. The mean of the changepoint component is centered at the most recent outcome (corresponding to an $\alpha_t$ of 1) and the mean of the non-changepoint component is an uncertainty $\tau_t$ weighted average of the most recent outcome and the mean of the prior predictive distribution over cannon location $c_t$ (corresponding to an $\alpha_t$ of $\tau_t$). A full derivation of the $\tau_t$ term has been provided in previous work (**Nassar et al., 2012**).

## Derivation of normative learning model for oddball condition

The generative process in the oddball condition can be defined in terms of the following sampling statements:

$$H = 0.125$$

$$S_t \sim Bernoulli(H)$$

$$C_t \sim Von\,Mises\left(C_{t-1},\,\sigma_{drift}\right)$$

$$B_t \mid S_t = 1 \sim Uniform(0, 359)$$

$$B_t \mid S_t = 0 \sim Von\,Mises(C_t,\,\sigma)$$

Once again, the inference problem is thus to infer the current cannon aim based on the sequence of observed cannonballs, which has a recursive solution that is similar to the solution of the changepoint condition described above:

$$
p(C_t|B_{1:t}) = \frac{\sum_{S_t}\int_{C_{t-1}} p(B_t \mid C_t, S_t)\, p(S_t)p(C_t \mid C_{t-1})p(C_{t-1}\mid B_{1:t-1})}{p(B_t \mid B_{1:t-1})}
\tag{7}
$$

Note that this equation differs from the changepoint solution only in that the likelihood of $B_t$ is now conditional on $S_t$ (now reflecting a binary oddball variable) whereas the $C_t$ is now conditionally independent of $S_t$. Once again, the equation can be expanded to reveal a mixture of two components:

$$p(C_t|B_{1:t}) = \frac{\int_{C_{t-1}} p(B_t, S_t = 1|C_t)p(C_t|C_{t-1})(H)p(C_{t-1}|B_{1:t-1}) + \frac{\int_{C_{t-1}} p(B_t|C_t, S_t = 0)p(C_t|C_{t-1})(1-H)p(C_{t-1}|B_{1:t-1})}{p(B_t|B_{1:t-1})}}{} \tag{8}$$

Where, once again, the first component reflects the 'oddball' predictive distribution and the latter reflects the 'non oddball' predictive distribution. As described for the changepoint condition, we could maintain this mixture distribution and propagate new mixture components with each observation (*Appendix 1—figure 2*, top). However, instead, we approximate the mixture distribution by replacing it with a Gaussian distribution that shares the same mean and variance as the full mixture (*Appendix 1—figure 2*, bottom). The mean of the updated mixture distribution can be written as the weighted sum of the two mixture components:

$$\hat{c}_{t+1} = (\hat{c}_{t+1}|s_t = 1)\Omega_t + (\hat{c}_{t+1}|s_t = 0)(1 - \Omega_t) \tag{9}$$

In the case of an oddball, the mean of the predictive distribution over cannon aim $(\hat{c}_{t+1}|s_t = 1)$ is equivalent to the prior predictive mean, as the likelihood distribution $(B_t|s_t = 1)$ is uniformly distributed for oddball trials. The mean of the non-oddball distribution is, like in the non-changepoint case in the changepoint condition, a weighted average of the prior predictive mean and the most recent outcome. This combination can also be expressed in terms of an error-driven learning rule:

$$
\begin{aligned}
\hat{c}_{t+1} &= (\hat{c}_{t+1}|s_t = 1)\Omega_t + (\hat{c}_{t+1}|s_t = 0)(1 - \Omega_t) \\
&= \hat{c}_t\Omega_t + (\hat{c}_t + \tau_t\delta_t)(1 - \Omega_t) \\
&= \hat{c}_t\Omega_t + \hat{c}_t - \hat{c}_t\Omega_t + \tau_t\delta_t - \Omega_t\tau_t\delta_t \\
&= \hat{c}_t\Omega_t - \hat{c}_t\Omega_t + \hat{c}_t + \tau_t\delta_t - \Omega_t\tau_t\delta_t \\
&= \hat{c}_t + \tau_t\delta_t - \Omega_t\tau_t\delta_t \\
&= \hat{c}_t + (\tau_t - \Omega_t\tau_t)\delta_t \\
&= \hat{c}_t + \alpha_t\delta_t
\end{aligned}
\tag{10}
$$

Note that the learning rate equation $\alpha_t := (\tau_t - \Omega_t\tau_t)$ differs from that in the changepoint condition in that it does not include a $(+\Omega_t)$ term. This is because the oddball mixture component is centered on the previous belief (as oddballs don't affect the position of the cannon), and thus higher levels of oddball probability $\Omega$ push learning rates towards zero, rather than towards one.

