## [Decision Letter]

Thank you for submitting your article "Statistical context dictates the relationship between feedback-related EEG signals and learning" for consideration by *eLife*. Your article has been reviewed by three peer reviewers, including Tobias H Donner as the Reviewing Editor and Reviewer #1, and the evaluation has been overseen by a Reviewing Editor and Timothy Behrens as the Senior Editor. The following individuals involved in review of your submission have agreed to reveal their identity: Jonas Obleser (Reviewer #2); Redmond G O'Connell (Reviewer #3).

The reviewers have discussed the reviews with one another and the Reviewing Editor has drafted this decision to help you prepare a revised submission.

Summary:

Nassar, Bruckner, and Frank examine the context-dependence of the impact of surprising sensory events on learning and choice behaviour. To this end, they have participants perform a continuous sensory decision-making task in two different statistical contexts: one in which mean of the process generating the evidence changes at unpredictable times ("change point task") and one in which this generative process does not change, but – by design – generates occasional outliers. Surprising evidence samples should elicit an adjustment of choice behavior in the first context, but be ignored in the second context.

Behavioural modelling shows that this normative prediction holds for human participants: They do factor surprising evidence samples into behaviour differently depending on the statistical context. The authors also examine EEG data for signatures of this context-dependent encoding of surprise and learning. A centro-parietal positivity in the evoked response scales with surprise irrespective of statistical context; but the influence of this response component on choice updating is conditional on context. The authors link the response component to the classic P3b or P300 of the EEG. The authors conclude that the P3 provides a general surprise signal, which is fed to a downstream process which translates surprise into a contextually appropriate behavioural adjustment.

This study addresses an interesting and timely question. It uses an original approach and is generally well executed. Specifically, all reviewers were impressed by the behavioral modelling part, but there are some issues pertaining the approach and interpretation of the EEG part, which should be resolved prior to publication.

Essential revisions:

1) Behavioral modelling.

Please present the normative model in more depth, in a longer methods section or supplement. Specifically, you should (i) derive the oddball version of the model, and (ii) explain the difference between the change-point and oddball versions of the model. A description of how the application of the model to a circular stimulus space differs from previous versions of the model would also be helpful.

2) Relationship to previous literature and theory on P3.

2a) Tease apart the novel aspects and the replication of established findings more explicitly.

The finding that P300 indexes surprise is already well-established in the oddball literature. For example, Kollossa et al., 2013 state: "It has long been recognized that fluctuations in P300 amplitude reflect the degree of surprise.". What is novel here seem to be two things: first, the establishment of the same surprise sensitivity in the context of a change point process; and second, the context-dependent relation of the signal to learning. This should be clarified throughout the manuscript, including the Impact Statement. Previous studies quantifying the link between P3 and surprise should be cited and discussed – specifically:

- Mars et al., (2008)

- Work by Bruno Kopp, e.g. Kopp et al., (2016).

2b) Relation to existing accounts of P3.

The discussion of how the present results relate to previous accounts of the P3 is somewhat confusing and should be clarified. In the Introduction the authors initially only allude to the context updating theory of the P3 but other accounts are mentioned in the Discussion section. A key issue here is that no clear initial hypotheses are derived from these models in the Introduction and the efforts to relate the present results to the models in the Discussion section is unclear – in several instances it is initially suggested that the present findings are at odds with a given theory but then acknowledged that the findings are potentially reconcilable. If the authors are unable to generate unique hypotheses for the present data based on previous theories of the P3, then this should be stated clearly.

Much of the P3 literature and the explanatory accounts have tended to centre on responses to stimuli that call for an immediate choice and report. Here the authors are effectively looking at feedback-related responses and it does seem difficult to generate specific predictions from the models in this particular context but that is not necessarily a limitation of the models, more a matter of unexplored territory requiring additional simulations and empirical research. For example, if the P3 reflects the perceptual decision relating to the location of the canonball (i.e. an evidence accumulation process) then, in line with the expectation-related bound modulations proposed in sequential sampling models, one would expect larger responses for less expected canonball locations and this would effectively fit the bill of a surprise response and would be expected to relate to conditional behaviour updating. Alternatively, and the peak timing of the P3 might speak to this, the P3 may largely/partly reflect the process of selecting the next shield position in light of the current outcome and more surprising events may prompt more careful deliberation resulting in high decision bounds and larger signal amplitudes. Our impression is that the present results are interesting in their own right but do not necessarily arbitrate among existing explanatory accounts of the P3.

In the final paragraphs the authors actually lay out a compelling account of what might be going on here without necessitating a full functional account of the P3: the P3 surprise response can play a role in triggering a change in the latent state. We suggest leading with this and following up with a discussion of the relevant theories.

2c) Relationship between P3 and the EEG component identified here.

While we appreciate the general "data-driven" approach used by the authors, we noticed that it inevitably raises questions about the relation to the so-called "P3 components" characterised in the oddball literature. This point requires more discussion.

3. Data analysis.

We believe that several aspects of the data analysis require further attention, specifically:

3a) The authors state that the critical PE*surprise*condition (or PE*EEG*condition) regressors indicate whether "surprise (EEG signal) tends to increase learning in the change-point condition but decrease learning in the oddball condition". But these interaction term regressors only test for a significant interaction – significant β weights do not imply a sign flip between conditions (increase in one condition, decrease in the other). For example, if surprise increased learning in the change point task, but does not correlate with learning in the oddball task, this might still yield a significant interaction. A sign flip should be assessed via posthoc comparisons.

3b) Subsection “Electrophysiological signatures of feedback processing”: With two conditions, oddball and changepoint, in this experiment, how can we have separate regressor weight estimates for both (one dummy variable coding condition would suffice/avoid collinearity)?

3c) The contrast "surprise" based on those two regressors might be modelled too liberally: A contrast setting both conditions to "1" is not necessarily identical to a true conjunction (i.e., both regressors driving the EEG significantly). This has been dealt with extensively in the fMRI/GLM literature. In short, outcomes from this "surprise" contrast are not necessarily as decisive as outcomes from a true difference contrast ("learning").

3c) Figure 5: Why should (behavioural) learning outcome be used to predict (on the y-axis) the temporally preceding positivity in the EEG? Also, the entire figure seems to stand on statistically shaky grounds, with p values in the.02-.04 range in highly sophisticated/convoluted models with many researcher degrees of freedom. Under the null, the result in Figure 5C would be as surprising (4 to 5 heads in row). The authors should do more to convince the reader that we are not looking at some lucky, highly selective results.

3d) The authors highlight an early frontocentral modulation in Figure 3 as being the P3a however the traces 3D indicate that this signal is equal in amplitude for expected and oddball stimuli. Shouldn't it be larger for oddballs if it is indeed a P3a?

3e) We suggest toning down the language in certain instances where the authors seem to imply that they have established a causal role for the P3 in belief updating e.g. Our findings are consistent with a number of studies that have suggested the P300 is related to surprise (9,14,17,24), but extend them by demonstrating the role of the signal in controlling the degree to which new information affects updated beliefs.

3f) The authors excluded 12/37 subjects excluded from EEG analysis because of low data quality. The criterion of excluding any subject with >25% artifactual trials seems rather stringent. Can you provide more rationale for the procedure? Are the main results robust with respect to such (arbitrary) selection criteria?

3g) 0.5 Hz is quite a severe high-pass cutoff and likely to attenuate some of the P3 activity. We don't think this could account for the significant effects of surprise etc but we would encourage the authors to repeat their key analyses with a substantially lower cutoff (e.g. 0.05 Hz) just to make sure that nothing changes

3f) What reference channel did the authors use for the EEG analyses – grand average?

---

## [Author Response]

Summary:Nassar, Bruckner, and Frank examine the context-dependence of the impact of surprising sensory events on learning and choice behaviour. To this end, they have participants perform a continuous sensory decision-making task in two different statistical contexts: one in which mean of the process generating the evidence changes at unpredictable times ("change point task") and one in which this generative process does not change, but – by design – generates occasional outliers. Surprising evidence samples should elicit an adjustment of choice behavior in the first context, but be ignored in the second context.Behavioural modelling shows that this normative prediction holds for human participants: They do factor surprising evidence samples into behaviour differently depending on the statistical context. The authors also examine EEG data for signatures of this context-dependent encoding of surprise and learning. A centro-parietal positivity in the evoked response scales with surprise irrespective of statistical context; but the influence of this response component on choice updating is conditional on context. The authors link the response component to the classic P3b or P300 of the EEG. The authors conclude that the P3 provides a general surprise signal, which is fed to a downstream process which translates surprise into a contextually appropriate behavioural adjustment.This study addresses an interesting and timely question. It uses an original approach and is generally well executed. Specifically, all reviewers were impressed by the behavioral modelling part, but there are some issues pertaining the approach and interpretation of the EEG part, which should be resolved prior to publication.Essential revisions:1) Behavioral modelling.Please present the normative model in more depth, in a longer methods section or supplement. Specifically, you should (i) derive the oddball version of the model, and (ii) explain the difference between the change-point and oddball versions of the model. A description of how the application of the model to a circular stimulus space differs from previous versions of the model would also be helpful.

We now include a full derivation of the normative learning model for the oddball condition in the supplementary material and refer interested readers to it:

“…leading to an error driven learning rule in which learning rate is adjusted dynamically from trial to trial, allowing us to derive normative prescriptions for learning for both conditions (see supplementary material for full derivation).”

We now also provide additional information about how the circular distributions were handled with respond to modeling and fitting the regression model as follows:

“Unlike standard regression in which the error distribution is assumed to be normal, our model imposed a circular (Von-Mises) distribution of errors around the predicted update. Maximum posterior coefficients for each individual subject were estimated using the fmincon optimization tool in Matlab (Mathworks, Natick, MA, USA) and t-tests were performed on the regression coefficients across participants to test for significant contributions of each term to update behavior. Weak Gaussian zero-centered priors were included to regularize coefficients of interest. In the purely behavioral analysis the width of priors over coefficients on standardized predictors was set to 5. In the analyses that included EEG predictors, EEG-based predictors were regularized using a zero centered Gaussian prior with a standard deviation of 0.1 (as compared to a standard deviation of 1 for the predicted learning rates from the behavioral model) making the regularization for the EEG terms stronger than that on the competing behavioral prediction term by a factor of 10 (thereby allowing preferential explanation of shared variance by the other terms in the model).”

We also plan to make the analysis and modeling code available on the first author’s website once the paper has been accepted in final form.

2) Relationship to previous literature and theory on P3.2a) Tease apart the novel aspects and the replication of established findings more explicitly.The finding that P300 indexes surprise is already well-established in the oddball literature. For example, Kollossa et al., 2013 state: "It has long been recognized that fluctuations in P300 amplitude reflect the degree of surprise.". What is novel here seem to be two things: first, the establishment of the same surprise sensitivity in the context of a change point process; and second, the context-dependent relation of the signal to learning. This should be clarified throughout the manuscript, including the Impact Statement. Previous studies quantifying the link between P3 and surprise should be cited and discussed – specifically:- Mars et al., (2008)- Work by Bruno Kopp, e.g. Kopp et al., (2016).

We now make this clear in the Impact statement:

“The P300, an EEG component known to be evoked by surprising events, predicts learning in a bidirectional manner that depends critically on the surrounding statistical context.”

And in the Introduction:

“While the central parietal component of the P300 (P3b) has been long known to reflect surprise (Mars et al., 2008; Kolossa, Kopp and Fingscheidt, 2015; Kopp et al., 2016; Seer et al., 2016; Kolossa et al., 2012), recent work suggests it relates to learning (Fischer and Ullsperger, 2013) even after controlling for the degree of surprise in changing environments (Jepma et al., 2016; Jepma et al., 2018).”

2b) Relation to existing accounts of P3.The discussion of how the present results relate to previous accounts of the P3 is somewhat confusing and should be clarified. In the Introduction the authors initially only allude to the context updating theory of the P3 but other accounts are mentioned in the Discussion section. A key issue here is that no clear initial hypotheses are derived from these models in the Introduction and the efforts to relate the present results to the models in the Discussion is unclear – in several instances it is initially suggested that the present findings are at odds with a given theory but then acknowledged that the findings are potentially reconcilable. If the authors are unable to generate unique hypotheses for the present data based on previous theories of the P3, then this should be stated clearly.Much of the P3 literature and the explanatory accounts have tended to centre on responses to stimuli that call for an immediate choice and report. Here the authors are effectively looking at feedback-related responses and it does seem difficult to generate specific predictions from the models in this particular context but that is not necessarily a limitation of the models, more a matter of unexplored territory requiring additional simulations and empirical research. For example, if the P3 reflects the perceptual decision relating to the location of the canonball (i.e. an evidence accumulation process) then, in line with the expectation-related bound modulations proposed in sequential sampling models, one would expect larger responses for less expected canonball locations and this would effectively fit the bill of a surprise response and would be expected to relate to conditional behaviour updating. Alternatively, and the peak timing of the P3 might speak to this, the P3 may largely/partly reflect the process of selecting the next shield position in light of the current outcome and more surprising events may prompt more careful deliberation resulting in high decision bounds and larger signal amplitudes. Our impression is that the present results are interesting in their own right but do not necessarily arbitrate among existing explanatory accounts of the P3.In the final paragraphs the authors actually lay out a compelling account of what might be going on here without necessitating a full functional account of the P3: the P3 surprise response can play a role in triggering a change in the latent state. We suggest leading with this and following up with a discussion of the relevant theories.

We agree with the reviewers that our results are interesting in their own right and do not arbitrate between existing theories of the P3. Thus, we have taken the suggested course of action and moved our discussion of theories of P3 to the end of the discussion, after our interpretation of the role of P3 signaling in our specific paradigm. We have also simplified this section, and made it clear that our results to not arbitrate between broader theories:

“Our findings are consistent with a number of studies that have demonstrated the P300 is related to surprise (Donchin, 1981; Wessel, 2016; Garrido et al., 2016; Jepma et al., 2016), but extend on them to reveal how the P300 relates to learning in different contexts. Our results are inconsistent with standard interpretations of the context updating theory of the P300 in which context is defined as a working memory for an observable stimulus (Donchin, 1981; Donchin and Coles, 2010; Polich, 2003; Polich, 2007), as under this definition a larger P300 should always lead to more learning. However, if the updated “contexts” were defined in terms of the latent states described above, the predictions of the context updating theory would indeed match our results. Thus, our results can constrain potential interpretations of the context updating theory, although they do not falsify the theory altogether. Nor do our results directly conflict with other prominent theories of P300 signaling including the idea that central parietal positivity might reflect accumulated evidence for a particular decision or course of action (Kelly and O'Connell, 2013; O'Connell, Dockree and Kelly 2012), or anticipate the need to inhibit responding (Wessel, 2016; Wessel and Aron, 2017), as both of these theories could be framed in terms of the latent states above (e.g., the accumulated evidence for a change in latent state or the need to inhibit responding until the appropriate latent state is loaded). Thus, our results do not arbitrate between these theories, but do require expansion of their interpretation (to include latent variables involved in the generation of outcomes) and also highlight their implications for learning when mechanistic interpretations are refined and applied to our task and data.”

2c) Relationship between P3 and the EEG component identified here.While we appreciate the general "data-driven" approach used by the authors, we noticed that it inevitably raises questions about the relation to the so-called "P3 components" characterised in the oddball literature. This point requires more discussion.

We now discuss the relationship between our purported P300 signals and those that have previously been characterized in the literature:

“We took a data-driven approach to identifying signals that related to surprising outcomes in different statistical contexts. The primary signal that we identified, however, was similar in timing (3D and F), location (3B), and sensitivity to surprise (3E and G) to those previously reported for the P300 (Kolossa et al., 2012; Kopp et al., 2016). The topography of our spatiotemporal cluster changed over time from frontocentral to centroparietal, consistent with inclusion of both an early frontocentral P3a component as well as a later centroparietal P3b component. Thus, although our methods were agnostic to detection of a specific signal, we interpret our results in the context of the larger literature relating to P300 signaling.”

3. Data analysis.We believe that several aspects of the data analysis require further attention, specifically:3a) The authors state that the critical PE*surprise*condition (or PE*EEG*condition) regressors indicate whether "surprise (EEG signal) tends to increase learning in the change-point condition but decrease learning in the oddball condition". But these interaction term regressors only test for a significant interaction – significant β weights do not imply a sign flip between conditions (increase in one condition, decrease in the other). For example, if surprise increased learning in the change point task, but does not correlate with learning in the oddball task, this might still yield a significant interaction. An sign flip should be assessed via posthoc comparisons.

We agree with the reviewers and now have performed the additional analysis that they recommended.

In terms of behavior, we have conducted an additional behavioral regression that includes separate PE*surprise terms for the changepoint condition (in which surprising trials reflect changepoints) and oddball condition (in which surprising trials reflect oddballs). We see that the changepoint coefficients are significantly positive across participants, whereas the oddball coefficients are significantly negative. We have created a supplementary figure (Figure 2—figure supplement 1) to report these results, and now include a reference to this figure in the main Results section.

To address the concern with our model of behavior that included the PE*condition*EEG interaction term, we have now performed another version of the same analysis where we include separate PE*EEG terms for each condition (changepoint, oddball). These terms include the mean-centered product for the modeled condition but are set to zero for the other condition. Applying this model to our updating behavior, we found that coefficients were positive for the changepoint condition, negative for the oddball condition, and significant in 3 of 4 cases when evaluated using the t-test method that we employ. It is also worth noting that the case where we did not see statistical significance (Early P300 cluster in the changepoint condition) the majority of participants showed an effect in the predicted direction (19/25, p-value for sign test for null hypothesis median = 0: 0.01), and thus we suspect that our inability to reject the null using a t-test was more related to the shape of the distribution than to a lack of effect. We now report this analysis in the Results section:

“Learning rate predictions derived from the regression model show that higher P300 signal strength predicts more learning in the changepoint condition (Figure 4E, orange), but less learning in the oddball condition (Figure 4E, blue) and this prediction was validated in a followup analysis that separately modeled the effect of EEG on learning in the changepoint and oddball conditions (figure 4—figure supplement 1).”

And display the results from it in figure 4—figure supplement 1.

3b) Subsection “Electrophysiological signatures of feedback processing”: With two conditions, oddball and changepoint, in this experiment, how can we have separate regressor weight estimates for both (one dummy variable coding condition would suffice/avoid collinearity)?

Apologies for the lack of clarity. All trials that did not involve a rare event (eg. neutral trials) were the implicit baseline to which our changepoint and oddball trials were compared. Thus, there were three types of trials – changepoints (rare events in the changepoint condition), oddballs (rare events in the oddball condition) and neutral trials (outcomes emerging from the expected transition). As the reviewers note, the trial type that we did not model explicitly (neutral trials) is captured by the intercept in our model. We have changed the description of our terms to make this point more clear:

“First we regressed feedback-locked EEG data collected simultaneously with task performance onto an explanatory matrix that included separate binary variables reflecting changepoint and oddball trials (as opposed to neutral trials that did not involve a rare event), amongst other terms (Figure 3A, left).”

3c) The contrast "surprise" based on those two regressors might be modelled too liberally: A contrast setting both conditions to "1" is not necessarily identical to a true conjunction (i.e., both regressors driving the EEG significantly). This has been dealt with extensively in the fMRI/GLM literature. In short, outcomes from this "surprise" contrast are not necessarily as decisive as outcomes from a true difference contrast ("learning").

We now directly examine the raw coefficients from which the surprise contrast was composed (changepoint and oddball) within the spatiotemporal cluster of interest and find that surprise is reflected similarly in the changepoint and oddball conditions. We have added a sentence to the Results section to describe this:

“In each cluster, changepoint and oddball trials contributed similarly to the overall surprise effect (Figure 3—figure supplement 1).”

And Figure 3—figure supplement 1 showing the raw coefficients.

3c) Figure 5: Why should (behavioural) learning outcome be used to predict (on the y-axis) the temporally preceding postivity in the EEG? Also, the entire figure seems to stand on statistically shaky grounds, with p values in the.02-.04 range in highly sophisticated/convoluted models with many researcher degrees of freedom. Under the null, the result in Figure 5C would be as surprising (4 to 5 heads in row). The authors should do more to convince the reader that we are not looking at some lucky, highly selective results.

Our intention in Figure 5 of our previous manuscript was to make two points:

1) individuals who conditionally modulate updating in response to surprise to a greater degree, also have P300 responses that conditionally predict updating to a greater degree. Or, more concisely, the P300 signal is most predictive of behavior in individuals that show the behavior we are trying to predict.

2) trial-to-trial variability in the P300 predicts trial-to-trial updating behavior beyond, even beyond what can be inferred using our behavioral model.

We now make these points in separate figures. In order to more clearly make the point about individual differences, we now include a panel in Figure 4 showing that the conditional learning EEG coefficients are positively correlated with our behavioral measure of the conditional effect of surprise on updating. This differs from our previous individual difference analysis in that it is based on the coefficients from our original EEG-based model of updates, rather than the one that includes an additional term to soak up variance that could be accounted for by our behavioral model (see below for updates to that analysis). We see a correlation between these variables that provides compelling evidence for a link between individual differences in behavior and EEG signaling (r = 0.54, p = 3 x 10^-4^).

We have also revised our analysis of the degree to which trial-to-trial EEG signals explain behavioral variability beyond that afforded by our original model. Our previous analysis was focused on two separate spatiotemporal clusters that were identified in our regression coefficient clustering procedure. However, after re-running our analysis using the reviewer suggestions (more inclusive high pass filter and eliminating subject exclusion criterion) we now identify one temporally extended spatiotemporal cluster. To more carefully test whether there is information in our cluster that can predict behavior, we now compute EEG signal strength in 40ms sliding windows of time across the duration of the P300 response. For each sliding window, we include the EEG signal strength in a linear model of participant updating behavior, and estimate coefficients that describe the degree to which the EEG signal directly predicts learning (direct learning) and conditionally predicts learning (conditional learning), while including the predicted updates of our behavioral model as a competing explanatory variable. We smooth coefficients over time and use cluster-based permutation testing to identify contiguous epochs over which coefficients deviated significantly from zero. We find that early in the P300 window (peak = 318 ms) conditional learning coefficients are positive (peak mean/SEM coefficient = 0.04/0.01; cluster corrected p value = 0.01), providing direct support for our claim that trial-to-trial variability in EEG signals can explain behavioral variability beyond what can be explained with behavioral measures alone.

3d) The authors highlight an early frontocentral modulation in Figure 3 as being the P3a however the traces 3D indicate that this signal is equal in amplitude for expected and oddball stimuli. Shouldn't it be larger for oddballs if it is indeed a P3a?

The early frontocentral modulation noted in Figure 3 is more positive for changepoint and oddball trials than for expected stimuli. This is indicated by the hot colors at frontocentral locations in Figure 3B (t-stat on CP+Oddball contrast) and is also evident in the positive event-related difference signals in the FCz channel [changepoint/oddball ERP – expected ERP] shown in figure 3e. Our inclusion of error bars (SEM) in figure 3D obscures this difference, as individual differences in the shape and magnitude of the ERP contribute substantially to the variance in the signal, and this was the motivation for including the error related difference signal on the right panels in which these components are removed.

3e) We suggest toning down the language in certain instances where the authors seem to imply that they have established a causal role for the P3 in belief updating e.g. Our findings are consistent with a number of studies that have suggested the P300 is related to surprise (9,14,17,24), but extend them by demonstrating the role of the signal in controlling the degree to which new information affects updated beliefs

We agree with the reviewers that our results do not unequivocally demonstrate a causal role for the P300 in controlling the degree to which new information affects updated beliefs. We have changed this sentence as follows:

“Our findings are consistent with a number of studies that have demonstrated the P300 is related to surprise (Donchin, 1981; Wessel, 2016; Garrido et al., 2016; Jepma et al., 2016), but extend them to reveal how the P300 differentially relates to learning in different contexts.”

3f) The authors excluded 12/37 subjects excluded from EEG analysis because of low data quality. The criterion of excluding any subject with >25% artifactual trials seems rather stringent. Can you provide more rationale for the procedure? Are the main results robust with respect to such (arbitrary) selection criteria?

The primary author had been advised by a technician to remove low signal-to-noise participants from EEG analysis – specifically using epoch rejections greater than 25% as a criterion. However, after a thorough search of the literature and consulting with several experienced EEG researchers, we realized that (1) there is no accepted approach for removing participants who are likely to have low signal-to-noise and (2) many (the majority?) of high quality papers in the field exclude very few participants, if any.

To assess the degree to which our subject exclusion affected our results, we reproduced our primary analyses using all possible rejection criterions. We found our primary result (that the P300-like EEG signals conditionally relates to learning) was apparent and statistically significant for all but the most conservative criterion values (e.g., removing all participants but 5). The size of the effects noticeably decreased for more liberal criterions – potentially consistent with the idea that some subjects were contributing more noise than signal – however effects were still reliable even when no subjects are excluded. However, performing the same robustness check on the residual analysis (the EEG-based regression in which we included a competing term in the model to soak up known sources of behavioral variability) yielded mixed results – with a wide range of criterion values over which the conditional learning effect was significant and in the appropriate direction, but with a number of criterions, including the most liberal case in which all subjects are included, unable to reject the null hypothesis (see Author response image 1).

In light of this lack of consistency, and with reproducibility and best practices in mind, we have decided to remove the exclusion criterion altogether (eg. include all EEG data). This is not a perfect solution – however we feel that this conservative approach is the lesser of two evils. As we reported above, inclusion of all participants contributed to a change in the clusters identified in our initial analysis, but our key conclusions remain unchanged.

**Author response image 1. respfig1:** Robustness check on exclusion crierion value. Top: proportion of good epochs for each participant. Middle/bottom: analysis results for different exclusion criteria. Lines/shading reflect mean/SEM conditional learning coefficients in the base model (middle) and the model that included predictions for our best behavioral model (bottom). Early (yellow) and late (green) clusters are plotted for all possible exclusion criterion values (abscissa) and significance of t-test on the average of the two clusters is indicated by pink points (blue points indicate lack of significance).

3g) 0.5 Hz is quite a severe high-pass cutoff and likely to attenuate some of the P3 activity. We don't think this could account for the significant effects of surprise etc but we would encourage the authors to repeat their key analyses with a substantially lower cutoff (e.g. 0.05 Hz) just to make sure that nothing changes

We have now re-analyzed the data with a lower cutoff (0.05) and find similar results. However, using this lower frequency cutoff changes the clusters formed in our EEG analysis slightly – a primary result being that the two clusters that we previously referred to as the “early” and “late” P300 cluster are merged into a single prolonged cluster. In addition, the magnitude of P300 effects is larger when using the lower cutoff. Both of these findings are consistent with the reviewer conjecture that some P3 activity might have been attenuated in earlier analyses. Thus, we have redone all analyses using this lower threshold.

3f) What reference channel did the authors use for the EEG analyses – grand average?

EEG data were collected in reference to CPz and re-referenced during pre-processing to the grand average. We now report this explicitly in the Materials and methods section:

**“**Data were collected using CPz as a reference channel and re-referenced to the grand mean for analysis”